# The burden of traumatic brain injury from low-energy falls among patients from 18 countries in the CENTER-TBI Registry: A comparative cohort study

**Fiona E. Lecky**[1,2]*, **Olubukola Otesile**[1], **Carl Marincowitz**[1], **Marek Majdan**[3], **Daan Nieboer**[4], **Hester F. Lingsma**[4], **Marc Maegele**[5], **Giuseppe Citerio**[6,7], **Nino Stocchetti**[8,9], **Ewout W. Steyerberg**[4,10], **David K. Menon**[11], **Andrew I. R. Maas**[12,13], **CENTER-TBI Participants and Investigators**[¶]

1 Centre for Urgent and Emergency Care Research, Health Services Research Section, School of Health and Related Research, University of Sheffield, Sheffield, United Kingdom, 2 Emergency Department, Salford Royal Hospital, Salford, United Kingdom, 3 Department of Public Health, University of Trnava, Trnava, Slovakia, 4 Department of Public Health, Erasmus University Medical Center, Rotterdam, Netherlands, 5 Institute for Research in Operative Medicine, Witten/Herdecke University, Köln, Germany, 6 Neurointensive Care, Azienda Socio Sanitaria Territoriale di Monza, Monza, Italy, 7 School of Medicine and Surgery, Università degli Studi di Milano–Bicocca, Milan, Italy, 8 Department of Pathophysiology and Transplantation, University of Milan, Milan, Italy, 9 Neuroscience Intensive Care Unit, Fondazione IRCCS Cà Granda Ospedale Maggiore Policlinico, Milan, Italy, 10 Department of Biomedical Data Sciences, Leiden University Medical Center, Leiden, Netherlands, 11 University of Cambridge, Addenbrooke's Hospital, Cambridge, United Kingdom, 12 Department of Neurosurgery, Antwerp University Hospital, Edegem, Belgium, 13 University of Antwerp, Edegem, Belgium

¶ Membership of CENTER-TBI Participants and Investigators is provided in the Acknowledgements.
* f.e.lecky@sheffield.ac.uk

**Data Availability Statement:** Data cannot be shared as the Consortium Agreement established between the CENTER TBI beneficiaries specifies the

## Abstract

### Background

Traumatic brain injury (TBI) is an important global public health burden, where those injured by high-energy transfer (e.g., road traffic collisions) are assumed to have more severe injury and are prioritised by emergency medical service trauma triage tools. However recent studies suggest an increasing TBI disease burden in older people injured through low-energy falls. We aimed to assess the prevalence of low-energy falls among patients presenting to hospital with TBI, and to compare their characteristics, care pathways, and outcomes to TBI caused by high-energy trauma.

### Methods and findings

We conducted a comparative cohort study utilising the CENTER-TBI (Collaborative European NeuroTrauma Effectiveness Research in TBI) Registry, which recorded patient demographics, injury, care pathway, and acute care outcome data in 56 acute trauma receiving hospitals across 18 countries (17 countries in Europe and Israel). Patients presenting with TBI and indications for computed tomography (CT) brain scan between 2014 to 2018 were purposively sampled. The main study outcomes were (i) the prevalence of low-energy falls

need for data access agreements with third parties. Proposals to access the study data, data dictionary, analytic code, and analysis scripts may be submitted online https://www.center-tbi.eu/data. Proposals are subject to review by the management committee. A Data Access Agreement is required, and all access must comply with regulatory restrictions imposed on the original study.

**Funding:** CENTER-TBI was supported by the European Union 7th Framework program (EC grant 602150), recipient A.I.R. Maas. Additional funding was obtained from the Hannelore Kohl Stiftung (Germany) - recipient A.I.R. Maas, from OneMind (USA) - recipient A.I.R. Maas and from Integra LifeSciences Corporation (USA) - recipient A.I.R. Maas. The funders had no role in study design, data collection and analysis, decision to publish, or preparation of the manuscript.

**Competing interests:** I have read the journal's policy and the authors of this manuscript have the following competing interests: DM reports grants, personal fees and non-financial support from GlaxoSmithKline, grants and personal fees from NeuroTrauma Sciences, personal fees from Pfizer Ltd, personal fees from PressuraNeuro, grants and personal fees from Lantmannen AB, grants and personal fees from Integra, outside the submitted work;DM is an Academic Editor on PLOS Medicine's editorial board. AM reports grants from European Union Framework 7, grants from Integra Life Sciences, grants from Hannelore Kohl Stiftung, grants from Neurotrauma Sciences, grants from BioDirection, during the conduct of the study; personal fees from PressuraNeuro, personal fees from Integra LifeSciences, personal fees from NeuroTrauma Sciences, outside the submitted work;. F L reports grants from EU Framework 7, during the conduct of the study; grants from Trauma Audit and Research Network, outside the submitted work; there are no other relationships or activities that could appear to have influenced the submitted work. There are no other author competing interests.

**Abbreviations:** AIS, Abbreviated Injury Scale; CT, computed tomography; ED, emergency department; EMS, emergency medical service; GCS, Glasgow Coma Scale; ICU, intensive care unit; ISS, Injury Severity Score; TBI, traumatic brain injury.

causing TBI within the overall cohort and (ii) comparisons of TBI patients injured by low-energy falls to TBI patients injured by high-energy transfer—in terms of demographic and injury characteristics, care pathways, and hospital mortality. In total, 22,782 eligible patients were enrolled, and study outcomes were analysed for 21,681 TBI patients with known injury mechanism; 40% (95% CI 39% to 41%) (8,622/21,681) of patients with TBI were injured by low-energy falls. Compared to 13,059 patients injured by high-energy transfer (HE cohort), the those injured through low-energy falls (LE cohort) were older (LE cohort, median 74 [IQR 56 to 84] years, versus HE cohort, median 42 [IQR 25 to 60] years; $p < 0.001$), more often female (LE cohort, 50% [95% CI 48% to 51%], versus HE cohort, 32% [95% CI 31% to 34%]; $p < 0.001$), more frequently taking pre-injury anticoagulants or/and platelet aggregation inhibitors (LE cohort, 44% [95% CI 42% to 45%], versus HE cohort, 13% [95% CI 11% to 14%]; $p < 0.001$), and less often presenting with moderately or severely impaired conscious level (LE cohort, 7.8% [95% CI 5.6% to 9.8%], versus HE cohort, 10% [95% CI 8.7% to 12%]; $p < 0.001$), but had similar in-hospital mortality (LE cohort, 6.3% [95% CI 4.2% to 8.3%], versus HE cohort, 7.0% [95% CI 5.3% to 8.6%]; $p = 0.83$). The CT brain scan traumatic abnormality rate was 3% lower in the LE cohort (LE cohort, 29% [95% CI 27% to 31%], versus HE cohort, 32% [95% CI 31% to 34%]; $p < 0.001$); individuals in the LE cohort were 50% less likely to receive critical care (LE cohort, 12% [95% CI 9.5% to 13%], versus HE cohort, 24% [95% CI 23% to 26%]; $p < 0.001$) or emergency interventions (LE cohort, 7.5% [95% CI 5.4% to 9.5%], versus HE cohort, 13% [95% CI 12% to 15%]; $p < 0.001$) than patients injured by high-energy transfer. The purposive sampling strategy and censorship of patient outcomes beyond hospital discharge are the main study limitations.

## Conclusions

We observed that patients sustaining TBI from low-energy falls are an important component of the TBI disease burden and a distinct demographic cohort; further, our findings suggest that energy transfer may not predict intracranial injury or acute care mortality in patients with TBI presenting to hospital. This suggests that factors beyond energy transfer level may be more relevant to prehospital and emergency department TBI triage in older people. A specific focus to improve prevention and care for patients sustaining TBI from low-energy falls is required.

---

### Author summary

#### Why was this study done?

- Traumatic brain injury (TBI) poses a huge global disease burden, considered to mainly result from high-energy transfer mechanisms such as road traffic collisions, sports, falls from a height, and interpersonal violence.

- People injured through low-energy transfer (ground- or low-level falls) are considered less likely to sustain significant TBI, so can be given lower priority for acute specialist care within emergency medical services (triage decisions).

- Recent multinational studies challenge these assumptions by identifying falls as an important TBI causal mechanism—but these studies seldom describe fall height.

- The lack of clarity concerning the low-energy TBI disease burden hampers effective prevention and clinical management.

## What did the researchers do and find?

- We studied 21,681 patients with TBI presenting to 56 hospital emergency departments across Europe and Israel using an efficient registry methodology enabling a real-world approach.

- We found that the 40% of patients with TBI who were injured through low-energy falls were significantly older, more likely to be female, and more likely to be taking pre-injury drugs that prevent blood clotting than patients with TBI sustained through high-energy transfer.

- Despite similar rates of significant injury on the CT brain scan and of dying in hospital, patients injured through low-energy falls were half as likely to receive critical care or emergency intervention compared to those injured by high-energy transfer.

## What do these findings mean?

- Low-energy falls contribute to a significant portion of the TBI disease burden, which will increase as the global population ages.

- In older people, the assumption that energy transfer predicts brain injury severity and threat to life appears to lack validity.

- Factors beyond energy transfer level may be more relevant to prehospital and emergency department TBI triage in older people. The appropriateness of providing less intensive acute hospital care after low-energy TBI requires further study.

- Reduction of TBI disease burden requires specific prevention and therapy initiatives targeted at low-energy TBI.

## Introduction

Traumatic brain injury (TBI) is a complex, imperfectly understood global disease [1], defined as 'an alteration in brain function, or other evidence of brain pathology, caused by an external force' [2]. This external force transfers mechanical energy to the brain to a degree that impairs its capacity for normal functioning, with the extent of injury varying with impact energy level [3–5]. Injury classification by energy transfer mechanism (high-energy transfer most commonly results from road traffic collisions, falling from a height, blunt assault, or contact sports while low-energy transfer results from low-level falls or those from a standing height [6–8]) has received broad attention in general trauma care and informs emergency medical service (EMS) on-scene trauma triage [3,8,9], with patients injured by high-energy transfer mechanisms being conveyed to higher levels of care within specialist trauma centres. High-energy transfer mechanisms are also the major focus of injury prevention and safety initiatives. This

prioritisation is underpinned by an assumption that high-energy transfer is more likely to result in tissue damage that is potentially life-threatening or life-altering. Hence, the TBI disease burden is traditionally attributed to high-energy transfer mechanisms, with patients with TBI historically being described in terms of the presenting Glasgow Coma Scale (GCS; measure of consciousness level) [10] and head CT findings [11]—using International Classification of Diseases codes [12], the Abbreviated Injury Scale (AIS) [13], or the Marshall classification system [14]—rather than by energy transfer level.

Recent studies challenge the paradigm of prioritising high-energy transfer mechanisms as the best strategy for reducing TBI disease burden. Falls are now the leading cause of TBI in Europe and other high-income countries—particularly in older people [7,15–20]. Not only are older people more likely to sustain a low-level fall, but age-related changes to the brain and blood vessels increase the likelihood of a consequent significant intracranial injury and decrease recovery potential. However, most published TBI series do not differentiate high- from low-energy-transfer falls. A specific focus on low-energy TBI disease burden is required, and the appropriateness of current injury prevention, EMS, and trauma centre triage priorities should be addressed.

The CENTER-TBI (Collaborative European NeuroTrauma Effectiveness Research in TBI) Registry aims to address the entire spectrum of TBI to increase our understanding of the most complex disease in the most complex organ [21–23]. It was established alongside core data collection, aiming to capture 'real world' data in large numbers of individuals to inform improvements in clinical management and injury prevention [24].

The aim of this study is to assess the TBI disease burden attributable to low-energy mechanisms by determining the prevalence of low-energy transfer as a causal mechanism for TBI and comparing—by causal energy level—the demographic, injury, acute care pathway, and early outcome characteristics of patients with TBI who presented to CENTER-TBI Registry recruiting hospitals in Europe and Israel.

## Methods

We established a prospective CENTER-TBI Registry in 56 participating centres across 18 countries (17 in Europe and Israel; S8 Table). The study enrolled patients presenting between 1 December 2014 and 31 January 2018 (S1 Fig). TBI patient advocacy groups contributed to study design and conduct through the study advisory board, facilitated by a dedicated section on the study website. We conducted a comparative cohort study between patients injured through low-energy falls and those injured by higher energy transfer mechanisms.

### Inclusion and exclusion criteria

The CENTER-TBI Registry included patients of all ages with a suspected or clinical diagnosis of TBI in whom computed tomography (CT) brain scan was conducted [23]. There were no exclusion criteria for recruitment. In particular, the registry included patients with pre-existing cognitive impairment such as dementia, severely injured patients who died in the emergency department (ED) resuscitation room prior to imaging, and patients presenting more than 24 hours after injury. These 3 groups were excluded from recruitment to the separate CENTER-TBI core cohort study of 4,509 patients within study centres. Recruitment to the core and registry were mutually exclusive [24].

### Data collection procedures for the CENTER-TBI Registry

Clinical data were collated over a 37-month period. No specific study interventions were performed. Study research teams were notified on arrival of eligible patients or identified these by

screening their sites' radiology imaging directories and ED records. Eligible patients' registry variables were collated from the clinical record after the patient had been discharged (or died), and were entered into the electronic case report form. This registry methodology enabled data to be collected in batches to increase efficiency and reduce cost. Sampling was purposive; training was provided that study teams should include a representative sample of patients with TBI presenting within specific 24-hour periods (with the selected 24-hour periods covering equally the different weekdays and seasons of the year). No patient identifiers were stored. The CENTER-TBI Registry data are stored on a secure database, hosted by the International Neuroinformatics Coordinating Facility in Stockholm, Sweden. Data curation was conducted by the CENTER-TBI Registry Work Package lead centre (FEL and OO).

## Case report form variables

We collected variables describing demographics (age and sex), pre-existing health status [25] including pre-injury anticoagulant and antiplatelet use, mechanism of injury, injury severity descriptors (GCS and AIS), presenting physiological vital signs (blood pressure, oxygen saturation, and pupillary responses), radiologically reported CT brain findings (classified as presence or absence of small or large epidural haematoma/acute subdural haematoma/brain contusion, subarachnoid haemorrhage, midline shift, basal cistern compression, and individual patient Marshall grading of CT findings [14]), processes of care (intubation status prior to arrival at study centre, method of referral to study centre [direct from scene versus secondary transfer from a referring hospital], time from study hospital arrival to CT brain scan, intensive care unit [ICU] admission, and key emergency interventions, i.e. craniotomy, intracranial pressure monitoring, decompressive craniectomy, external limb fixation, emergency laparotomy, thoracotomy, and extraperitoneal pelvic packing), and immediate outcome of care in terms of hospital mortality, length of stay (or time to death, where this occurred), and destination on discharge (own home, nursing home, another hospital, or rehabilitation centre). These variables were derived from the Utstein trauma template used for standard trauma registry collection across Europe, North America, and Australasia [26]. Patients were described in terms of 1 of 3 clinical care pathways after presentation, triage, and CT brain scan: discharge home or direct to the mortuary from the ED, admission to the hospital but not to the ICU, or admission to the ICU. A web-based data entry format was implemented.

## Classification of patients by injury energy mechanism

TBI patients with the following mechanisms of injury were classified as having high-energy TBI: motor vehicle incidents, collisions involving bicycles and motorcycles, falls from a height, assaults, sports, and other high-energy transfer incidents; falls from a standing or low height were classified as low-energy TBI [6–8].

## Statistical analysis

Increased understanding of TBI disease burden through improved classification, and identification of effective care, are core objectives in the 2015 CENTER-TBI published protocol of the core and registry studies [23]. The protocol also specified the patient characteristic, care pathway, and outcome variables for collection within the registry that could best support these objectives [23]. The study objectives and analysis plan for determining the low-energy TBI disease burden were stimulated by publications [3,6,7,18] highlighting falls as an increasingly common mechanism causing TBI during and shortly after the completion of CENTER-TBI patient recruitment. The analysis plan was agreed on by the authors at a study meeting at the University of Antwerp in January 2019. We prespecified the energy transfer patient

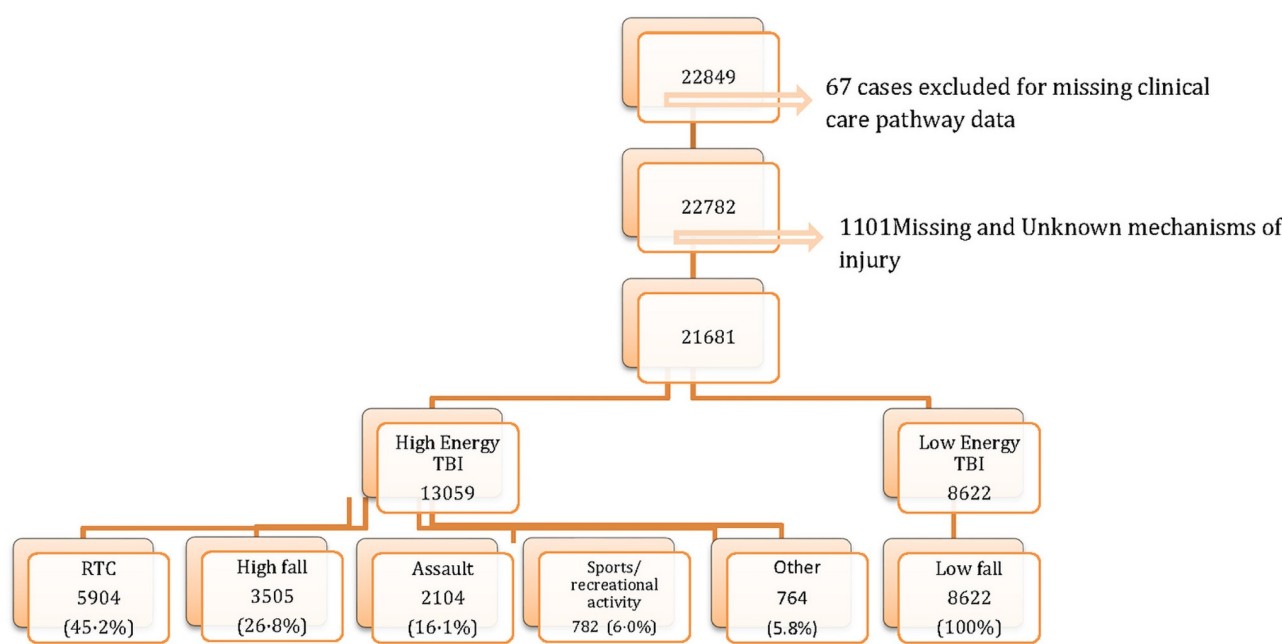

**Fig 1. Identification of patients injured by high- and low-energy transfer.** RTC, road traffic collision; TBI, traumatic brain injury.

classification—informed by the literature (Fig 1); variables and statistical methods for comparison by energy transfer level are available in S1 Analysis Plan. The study analysis plan has not been modified since. The analysis was based on the CENTER-TBI Registry data version 2.0, downloaded from a data management tool, Neurobot (https://center-tbi.incf.org/).

We carried out comparative analysis by energy transfer level. Continuous and ordinal variables (age, time intervals, GCS, AIS [10,13], and Injury Severity Score (ISS) [27]) are presented as median and interquartile range (IQR), while categorical variables are presented as number and percentage. Chi-squared tests were used to compare categorical variables between low- and high-energy TBI categories, while non-parametric continuous and ordinal variables were compared using the Mann–Whitney test.

Analyses were performed using IBM Statistical Package for Social Sciences (SPSS) version 23, Microsoft Excel 2010, and RStudio (version 1.0.136).

An Excel radar plot compared the time of day of hospital arrival by energy transfer level; the arrival times of the overall cohort were also compared to those of patients with TBI submitted to the largest European trauma registry—the Trauma Audit and Research Network (https://www.tarn.ac.uk), which has Section 251 (Health Research Authority) approval for analysis of anonymised data. This facilitated appraisal of the purposive sampling strategy.

## Ethics statement

The CENTER-TBI study (EC grant 602150) was conducted in accordance with all relevant laws of the European Union if directly applicable or of direct effect and all relevant laws of the country where the recruiting sites were located, including, but not limited to, relevant privacy and data protection laws and regulations, relevant laws and regulations on the use of human materials, and all relevant guidance relating to clinical studies at various times in force, including, but not limited to, the ICH Harmonised Tripartite Guideline for Good Clinical Practice (CPMP/ICH/135/95) and the World Medical Association Declaration of Helsinki.

Informed consent was not required as only administrative and routinely collected clinical data were accessed. However, national and local institutional review board approvals were obtained as per national guidelines. For example, within the UK, approval was obtained from the Health Research Authority.

Ethical approval was obtained for each recruiting site. The list of sites, ethical committees, approval numbers, and approval dates can be found at https://www.center-tbi.eu/project/ethical-approval.

This study is reported as per the Strengthening the Reporting of Observational Studies in Epidemiology (STROBE) guideline (S1 STROBE Checklist).

## Results

Fifty-six study centres from 18 countries (17 European countries and Israel; S1 Table) enrolled 22,849 patients to the CENTER-TBI Registry (for cumulative recruitment over the study, see S1 Fig). A total of 21,681 TBI patients with known clinical care pathway and injury mechanism were included in the overall cohort for analysis (Fig 1), a median of 247 (IQR 63 to 473) from each centre.

### Overall cohort

Patients enrolled into the CENTER-TBI Registry had a median age of 55 years (IQR 32 to 75 years) and a 61% (95% CI 60% to 62%) male preponderance; 55% (95% CI 54% to 56%) had pre-existing medical conditions, 12% (95% CI 11% to 13%) ($n$ = 2,578) and 11% (95% CI 10% to 13%) ($n$ = 2,466) were taking pre-injury anticoagulant or antiplatelet therapy. Overall, 82% (95% CI 81% to 82%) ($n$ = 17,702) presented to the study hospital ED with mild TBI (GCS 13–15) (Table 1), with 89% (95% CI 89% to 90%) having normal pupillary responses. Overall, 8.4% (95% CI 7.1% to 9.6%) ($n$ = 1,812) of patients arrived intubated, and 12% (95% CI 11% to 14%) (2,692) arrived as a result of a secondary transfer from a referring hospital. The radar plot of hospital arrival times demonstrated a post meridiem (PM) hospital arrival predominance within the overall study cohort similar to that of TBI patients submitted to the Trauma Audit and Research Network (S2 Fig). Patients presenting directly had CT brain imaging conducted a median (IQR) of 68 (34 to 141) minutes after ED arrival. The majority (57% [95% CI 56% to 58%]) were admitted to hospital, and 19% (95% CI 18% to 20%) to intensive care (an average of 2.5 hours after ED arrival). Overall, 31% (95% CI 30% to 32%) ($n$ = 6,746) of patients had injuries on CT brain scan. The most common of these were subarachnoid haemorrhage, small cerebral contusion, and small subdural haematoma (Tables 2 and 3). Extracranial injuries were usually present in moderate severity (median extracranial ISS = 4), and 11% (95% CI 9.8% to 12%) of patients received key emergency interventions, with craniotomy (3.5% [95% CI 2.2% to 4.8%]) within 3 hours of ED arrival being the most common. Hospital mortality was 6.7%, (95% CI 5.4% to 8.0%), with 71% (95% CI 70% to 72%) of patients being discharged directly to their own home (Table 4).

### Comparisons by energy transfer mechanism

**Patient characteristics and care pathway.** Forty percent (95% CI 39% to 41%) (8,622) of the overall cohort were injured as a consequence of low-energy falls. Detailed comparisons of patient characteristics between patients injured by low-energy falls (LE cohort) versus those injured by high-energy transfer mechanisms (HE cohort) are provided in Tables 1 and 2. Road traffic collisions, falls from a height, and assaults were the predominant causal mechanisms in patients injured by high-energy transfer (Fig 1); the prevalence of low-energy transfer varied from 30% to 50% in most participating countries and recruiting centres (Figs 2 and S1).

**Table 1. Demographics, injury mechanism, comorbidity, and presenting physiology of 21,681 TBI patients enrolled in the CENTER-TBI Registry.**

| Characteristic | Overall | High-energy TBI | Low-energy TBI | *p*-Value |
|---|---|---|---|---|
| Total | 21,681 | 13,059 (60.2) | 8,622 (39.8) | |
| **Demographic characteristics** | | | | |
| Age (years), median (IQR) | 55 (32–75) | 42 (25–60) | 74 (56–84) | <0.001[a] |
| Age | | | | |
| Under 16 years | 579 (2.7) | 488 (3.7) | 91 (1.1) | <0.001[b] |
| 16–64 years | 12,782 (59.0) | 9,934 (76.1) | 2,848 (33.0) | |
| 65 years and over | 8,317 (38.4) | 2,635 (20.2) | 5,682 (65.9) | |
| Male | | | | |
| Overall | 13,186 (60.8) | 8,833 (67.6) | 4,353 (50.5) | <0.001[b] |
| Under 16 years | 371 (64.1) | 316 (64.8) | 55 (60.4) | <0.001[b] |
| 16–64 years | 8,831 (69.1) | 7,044 (70.9) | 1,787 (62.7) | |
| 65 years and over | 3,982 (47.9) | 1,472 (55.9) | 2,510 (44.2) | |
| **Place of injury** | | | | |
| Street/highway | 7,324 (33.8) | 6,328 (48.5) | 996 (11.6) | <0.001[b] |
| Home/domestic | 8,295 (38.3) | 2,860 (21.9) | 5,435 (63.0) | |
| Work/school | 746 (3.4) | 587 (4.5) | 159 (1.8) | |
| Sport/recreational | 721 (3.3) | 671 (5.1) | 50 (0.6) | |
| Public location | 3,394 (15.7) | 2,000 (15.3) | 1,394 (16.2) | |
| Other | 735 (3.4) | 308 (2.4) | 427 (5.0) | |
| Unknown | 448 (2.1) | 292 (2.2) | 156 (1.8) | |
| Missing | 18 (0.1) | 13 (0.1) | 5 (0.1) | |
| **Pre-injury health status and medical history** | | | | |
| Normal healthy patient | 8,688 (40.1) | 7,285 (55.8) | 1,403 (16.3) | <0.001[b] |
| Mild systemic disease | 6,266 (28.9) | 3,391 (26.0) | 2,875 (33.3) | |
| Severe systemic disease | 5,105 (23.5) | 1,569 (12.0) | 3,536 (41.0) | |
| Life-threatening disease | 523 (2.4) | 116 (0.9) | 407 (4.7) | |
| Taking anticoagulants | 2,578 (11.9) | 744 (5.7) | 1,834 (21.3) | <0.001[b] |
| Taking platelet aggregate inhibitors | 2,466 (11.4) | 783 (6.0) | 1,683 (19.5) | <0.001[b] |
| Taking both anticoagulants and platelet aggregate inhibitors | 370 (1.7) | 120 (0.9) | 250 (2.9) | <0.001[b] |
| Intracranial lesions and taking anticoagulants | 580 (11.7) | 207 (6.5) | 373 (20.9) | <0.001[b] |
| Intracranial lesions and taking platelet aggregate inhibitors | 679 (13.7) | 269 (8.5) | 410 (23.0) | <0.001[b] |
| Intracranial lesions and taking both anticoagulants and platelet aggregate inhibitors | 114 (2.3) | 37 (1.2) | 77 (4.3) | <0.001[b] |
| **ED arrival physiology** | | | | |
| GCS, median (IQR)[c] | 15 (14–15) | 15 (14–15) | 15 (14–15) | <0.001[a] |
| GCS | | | | |
| Mild TBI (GCS 13–15) | 17,702 (81.6) | 10,270 (78.6) | 7,432 (86.2) | |
| Moderate TBI (GCS 9–12) | 830 (3.8) | 480 (3.7) | 350 (4.1) | |
| Severe TBI (GCS 3–8) | 1,195 (5.5) | 879 (6.7) | 316 (3.7) | |
| No sum | 1,835 (8.5) | 1,366 (10.5) | 469 (5.4) | |
| Hypoxia[c] | 287 (1.3) | 183 (1.4) | 104 (1.2) | 0.219[b] |
| Hypotension[c] | 434 (2.0) | 322 (2.5) | 112 (1.3) | <0.001[b] |
| Pupillary reactivity[c] | | | | <0.001[b] |
| Neither reacting | 552 (2.5) | 419 (3.2) | 133 (1.5) | |
| One reacting | 483 (2.2) | 313 (2.4) | 170 (2.0) | |
| Both reacting | 19,409 (89.5) | 11,694 (89.5) | 7,715 (89.5) | |

Data are *n* (%) unless otherwise indicated.

[a]By Mann–Whitney test for non-parametric variable distributions.

[b]By chi-squared test.

[c]Missing data percentage is 5.0%–10% for low- versus high-energy TBI comparison.

ED, emergency department; GCS, Glasgow Coma Score; IQR, interquartile range; TBI, traumatic brain injury.

**Table 2. Imaging findings, injury severity, therapeutic interventions, and care pathways of 21,681 TBI patients enrolled in the CENTER-TBI Registry.**

| Outcome | Overall | High-energy TBI | Low-energy TBI | p-Value |
|---|---|---|---|---|
| **Care pathway** | | | | |
| Emergency department | 9,286 (42.8) | 5,511 (42.2) | 3,775 (43.8) | 0.021[a] |
| Admission | 8,224 (37.9) | 4,365 (33.4) | 3,859 (44.8) | <0.001[a] |
| Intensive care unit | 4,171 (19.2) | 3,183 (24.4) | 988 (11.5) | <0.001[a] |
| **CT characteristics** | | | | |
| Abnormal CT | 6,746 (31.1) | 4,226 (32.4) | 2,520 (29.2) | <0.001[a] |
| Abnormal CT with intracranial lesion[b] | 4,959 (73.5) | 3,177 (75.2) | 1,782 (70.7) | <0.001[a] |
| EDH—small[b] | 619 (9.2) | 456 (10.8) | 163 (6.5) | <0.001[a] |
| EDH—large[b] | 239 (3.5) | 175 (4.1) | 64 (2.5) | 0.001[a] |
| ASDH—small[b] | 2,036 (30.2) | 1,271 (30.1) | 765 (30.4) | 0.808[a] |
| ASDH—large[b] | 983 (14.6) | 520 (12.3) | 463 (18.4) | <0.001[a] |
| Contusions—small[b] | 2,449 (36.3) | 1,698 (40.2) | 751 (29.8) | <0.001[a] |
| Contusions—large[b] | 567 (8.4) | 404 (9.6) | 163 (6.5) | <0.001[a] |
| Compressed basal cisterns[b] | 791 (11.7) | 548 (13.0) | 243 (9.6) | <0.001[a] |
| Midline shift[b] | 1,592 (23.6) | 883 (20.9) | 709 (28.2) | <0.001[a] |
| Subarachnoid haemorrhage[b] | 3,509 (52.0) | 2,397 (56.7) | 1,112 (44.1) | <0.001[a] |
| Marshall grading | | | | <0.001[a] |
| II—cisterns present with midline shift 0–5 mm and/or lesions present (high/mixed density < 25 cm); may include bone fragments | 14,743 (68.6) | 3,141 (24.3) | 1,761 (20.5) | |
| III—diffuse injury; cisterns compressed/absent with midline shift 0–5 mm (high/mixed density < 25 cm) | 4,902 (22.8) | 181 (1.4) | 36 (0.4) | |
| IV—diffuse injury; midline shift > 5 mm (high/mixed density < 25 cm) | 217 (1.0) | 107 (0.8) | 94 (1.1) | |
| V—surgically evacuated mass lesion | 201 (0.9) | 396 (3.0) | 363 (4.2) | |
| VI—non-evacuated mass lesion > 25 cm | 759 (3.5) | 401 (3.1) | 266 (3.1) | |
| Head/neck AIS, median (IQR) | 2 (1–3) | 2 (1–3) | 1 (1–2) | <0.001[c] |
| Cervical spine AIS, median (IQR) | 0 (0–0) | 0 (0–0) | 0 (0–0) | <0.001[d] |
| ISS, median (IQR) | 9 (4–17) | 9 (4–20) | 6 (3–12) | <0.001[c] |
| Extracranial ISS, median (IQR) | 4 (1–9) | 5 (1–13) | 2 (0–8) | <0.001[c] |
| **Processes of care** | | | | |
| Secondary referral—arrived from another hospital | 2,692 (12.4) | 1,720 (13.2) | 972 (11.3) | <0.001[a] |
| Length of stay (hours), median (IQR) | 17 (3–137) | 17 (3–137) | 19 (4–137) | 0.080 |
| Arrived intubated | 1,812 (8.4) | 1,496 (11.5) | 316 (3.7) | <0.001[a] |
| Key interventions | | | | |
| At least 1 key emergency intervention | 2,397 (11.1) | 1,752 (13.4) | 645 (7.5) | <0.001[a] |
| Craniotomy | 767 (3.5) | 401 (3.1) | 366 (4.2) | <0.001[a] |
| ICP monitor insertion | 753 (3.5) | 609 (4.7) | 144 (1.7) | <0.001[a] |
| Decompressive craniectomy | 275 (1.3) | 201 (1.5) | 74 (0.9) | <0.001[a] |
| External limb fixation | 293 (1.4) | 276 (2.1) | 17 (0.2) | <0.001[a] |
| Other[e] | 445 (2.1) | 369 (2.8) | 76 (0.9) | |

Data are n (%) unless otherwise indicated.

[a]Low- versus high-energy TBI comparison by chi-squared test.

[c]By Mann–Whitney test for non-parametric variable distributions.

[b]Denominator is those with abnormal CT.

[d]Cervical spine injury present in 21.0% and 13.3% of high- and low-energy TBI cohorts, respectively.

[e]Includes external ventricular drainage, interventional radiology, damage control thoracotomy and laparotomy, and extraperitoneal pelvic packing.

AIS, Abbreviated Injury Scale; ASDH, acute subdural haematoma; CT, computed tomography; EDH, extradural haematoma; ICP, intracranial pressure; IQR, interquartile range; ISS, Injury Severity Score; TBI, traumatic brain injury.

**Table 3. Median (interquartile range) times (in minutes) from arrival to imaging and key emergency interventions in study hospital (21,681 patients).**

| Time interval | Overall | High-energy TBI | Low-energy TBI | *p*-Value[a] |
|---|---|---|---|---|
| Time to CT in non-transferred patients | 68 (34–141) | 54 (29–109) | 99 (49–179) | <0.001 |
| Time to ICU admission in non-transferred patients | 151 (70–276) | 152 (74–270) | 150 (49–304) | 0.632 |
| Time to ICP monitor insertion | 197 (87–485) | 195 (89–472) | 216 (77–568) | 0.649 |
| Time to craniotomy | 142 (62–410) | 115 (60–286) | 190 (68–660) | <0.001 |
| Time to decompressive craniectomy | 165 (71–761) | 129 (70–657) | 235 (83–1,105) | 0.098 |
| Time to first extracranial emergency intervention | 137 (59–284) | 128 (59–267) | 170 (60–717) | 0.064 |

[a]From Mann–Whitney comparisons of non-parametric variable distributions—high- versus low-energy TBI.

CT, computed tomography; ICP, intracranial pressure; ICU, intensive care unit; TBI, traumatic brain injury.

Patients sustaining TBI from low-energy falls were significantly older (median [IQR]: LE cohort, 74 [56 to 84] years, versus HE cohort, 42 [25 to 60] years; *p* < 0.001), with 66% (95% CI 65% to 67%) aged 65 years and over, and were less likely to be male (LE cohort, 51% [95% CI 49% to 52%], versus HE cohort, 67% [95% CI 66% to 69%]; *p* < 0.001) than patients injured by high-energy mechanisms. Patients injured by low-energy mechanisms were more likely to be injured at home (Table 1) and to arrive at the ED during daylight hours (S2 Fig). The low-energy TBI cohort had a significantly higher prevalence of pre-existing disease (LE cohort, 79% [95% CI 78% to 80%], versus HE cohort, 39% [95% CI 38% to 40%]; *p* < 0.001) and sole anticoagulant (LE cohort, 21% [95% CI 19% to 23%], versus HE cohort, 5.7% [95% CI 4.0% to 7.3%]; *p* < 0.001) or antiplatelet (LE cohort, 20% [95% CI 18% to 21%], versus HE cohort, 6.0% [95% CI 4.3% to 7.6%]; *p* < 0.001) usage than the high-energy TBI cohort, but were more likely to present with mild TBI (GCS 13–15; LE cohort, 86% [95% CI 85% to 87%], versus HE cohort, 79% [95% CI 78% to 79%]) and with normal pupils and vital signs (Table 1) than patients injured by high-energy mechanisms. These differences persisted within care pathways (Fig 3; S2 Table).

Patients injured by low-energy falls were less likely to present to hospital intubated (LE cohort, 3.7 [95% CI 1.6% to 5.7%], versus HE cohort, 11% [95% CI 9.8% to 13.0%]; *p* < 0.001) or via a secondary transfer (LE cohort, 11% [95% CI 9.2% to 13.0%], versus HE cohort, 13% [95% CI 12% to 14%]; *p* < 0.001) and were more likely to have a delayed CT brain scan (median [IQR]: LE cohort, 99 [49 to 179] minutes after ED arrival, versus HE cohort, 54 [29 to 109] minutes; *p* < 0.001). The proportion of the LE cohort admitted to hospital was similar to that of the HE cohort (LE cohort, 56% [95% CI 55% to 58%], versus HE cohort, 58% [95% CI 57% to 59%]; *p* = 0.021), but the proportion receiving ICU care was half that of high-energy TBI patients (LE cohort, 11% [95% CI 9.4% to 13%], versus HE cohort, 24% [95% CI 23% to

**Table 4. Hospital mortality and discharge destination from study hospital (21,681 patients).**

| Outcome | Overall | High-energy TBI | Low-energy TBI | *p*-Value[a] |
|---|---|---|---|---|
| Total | 21,681 | 13,059 (60.2) | 8,622 (39.8) | |
| Hospital mortality | 1,453 (6.7) | 913 (7.0) | 540 (6.3) | 0.825 |
| Discharged home | 15,324 (70.7) | 9,458 (72.4) | 5,866 (68.0) | <0.001 |
| Discharged to other hospital | 2,151 (9.9) | 1,351 (10.3) | 800 (9.3) | 0.010 |
| Discharged to rehabilitation | 1,221 (5.6) | 776 (5.9) | 445 (5.2) | 0.015 |
| Discharged to nursing home | 1,122 (5.2) | 289 (2.2) | 833 (9.7) | <0.001 |

Data are *n* (%).

[a]Low- versus high-energy TBI comparison by chi-squared test.

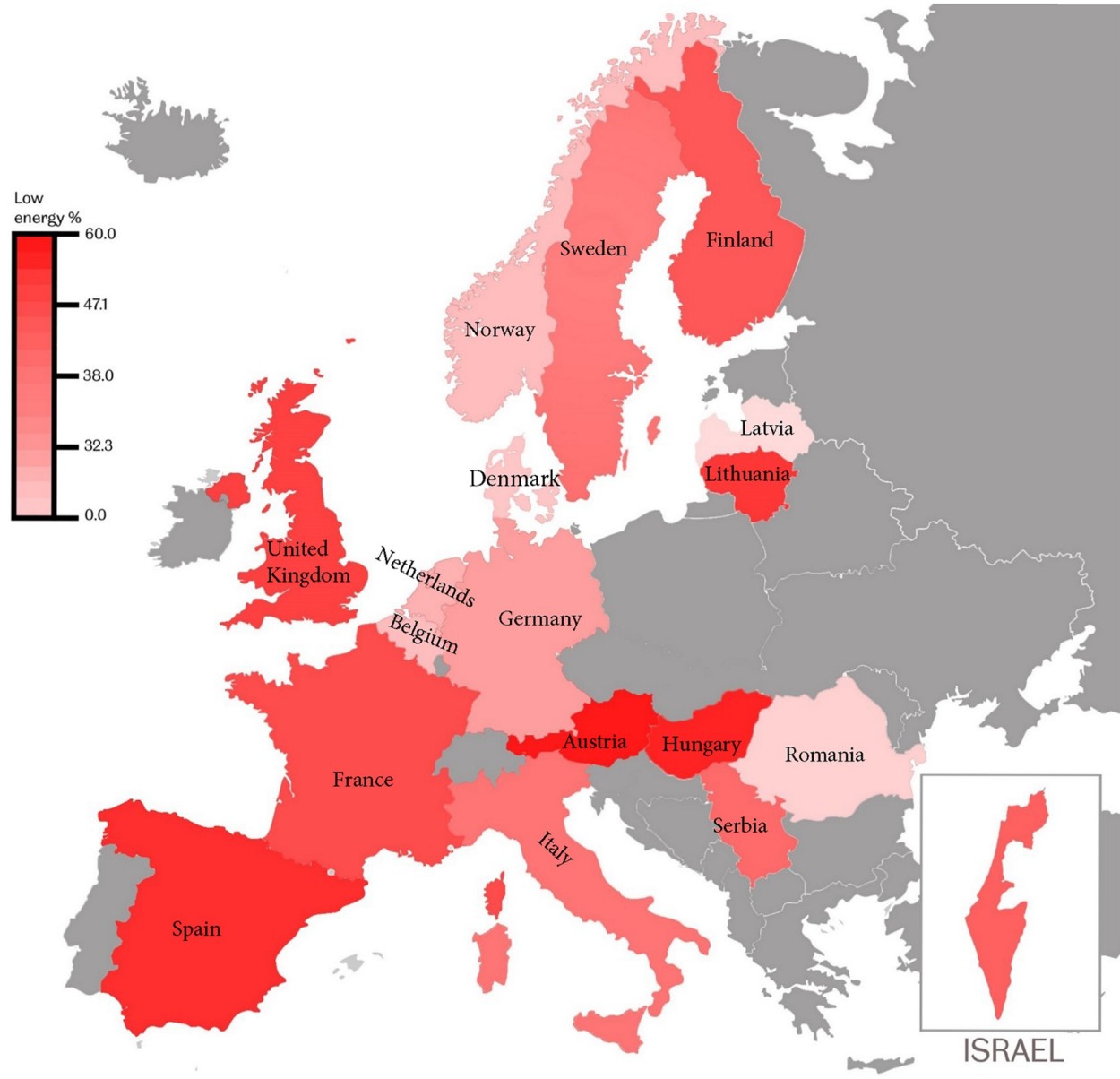

**Fig 2. Prevalence of low-energy transfer as TBI causal mechanism in countries participating in the CENTER-TBI Registry.** Colour shading as per key. Map from https://en.wikipedia.org/wiki/File:Europe_blank_map.png.

26%]; $p < 0.001$), although times from arrival to ICU admission were similar. Patients in the low-energy TBI cohort had a similar hospital length of stay (median [IQR]: 19 [4 to 137] hours) to the high-energy TBI cohort (Tables 2 and 3).

**Injury characteristics.** In patients injured by low-energy falls, the proportion with abnormalities detected on CT scan (including skull fracture) (29% [95% CI 27% to 31%]) and the proportion of abnormal scans showing intracranial injury (70% [95% CI 69% to 73%]) were lower than those of patients injured by high-energy mechanisms (32% [95% CI 31% to 34%] and 75% [95% CI 74% to 77%], respectively; $p < 0.001$). Patients with intracranial injuries sustained through low-energy mechanisms were more likely to be taking anticoagulants, platelet

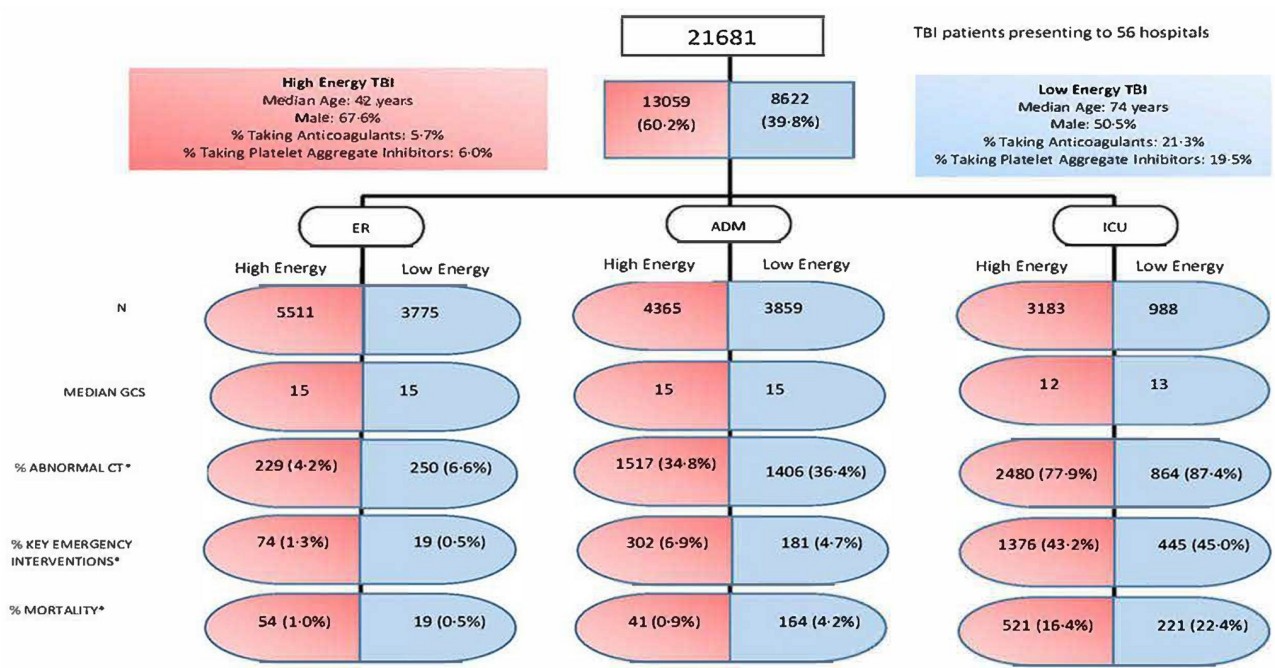

**Fig 3. High-energy and low-energy TBI patient characteristics compared within 3 care pathways.** The 3 care pathways are as follows: discharged or died in emergency department (ER; left column), admitted to ward but not receiving critical care in study hospital (ADM; central column), and admitted and received critical care in study hospital intensive care unit (ICU; right column). *Denominator is number of patients in energy transfer and care pathway cohort. CT, computed tomography; GCS, Glasgow Coma Score; TBI, traumatic brain injury.

aggregate inhibitors, or both (LE cohort, 21% [95% CI 17% to 25%], 23% [95% CI 19% to 27%], 4.3% [95% CI 0.0% to 8.8%], respectively, versus HE cohort, 6.5% [95% CI 3.2% to 9.9%], 8.5% [95% CI 5.1% to 12%], and 1.1% [95% CI 0.0% to 4.6%], respectively; $p < 0.001$). Rates of large acute subdural haematoma (LE cohort, 18% [95% CI 15% to 22%], versus HE cohort, 12% [95% CI 9% to 15%]; $p < 0.001$) and presence of midline shift (LE cohort, 28% [95% CI 25% to 31%], versus HE cohort, 21% [95% CI 18% to 24%]; $p < 0.001$)—as a proportion of patients with abnormalities on CT brain scan—were higher in the low-energy TBI cohort. However patients with low-energy injuries were significantly less likely to have small epidural haemorrhage (LE cohort, 6.5% [95% CI 2.7% to 10%], versus HE cohort, 11% [95% CI 7.9% to 14%]; $p < 0.001$), large epidural haemorrhage (LE cohort, 2.5% [95% CI 0.0% to 6.3%], versus HE cohort, 4.1% [95% CI 1.2% to 7.0%]; $p < 0.001$), small contusions (LE cohort, 30% [95% CI 27% to 33%], versus HE cohort, 40% [95% CI 38% to 43%]; $p < 0.001$), large contusions (LE cohort, 6.5% [95% CI 2.7% to 10%], versus HE cohort, 9.6% [95% CI 6.7% to 12.4%]; $p < 0.001$), subarachnoid haemorrhage (LE cohort, 44% [95% CI 41% to 47%], versus HE cohort, 57% [95% CI 55% to 59%]; $p < 0.001$), and basal cistern compression (LE cohort, 9.6% [95% CI 5.9% to 13.0%], versus HE cohort, 13% [95% CI 10% to 16%]; $p < 0.001$). They also had less severe extracranial injuries (median [IQR] extracranial ISS: LE cohort, 2 [0 to 8], versus HE cohort, 5 [1 to 13]; $p < 0.001$). The rate of small subdural haematoma in low-energy TBI patients was similar to that in high-energy TBI patients (30% [95% CI 27% to 33%]) (Table 2). The greatest relevant differences in Marshall CT grading were in rates of diffuse injury (III and IV; Table 2); diffuse injuries were 50% more common in the high-energy TBI cohort. Overall, the low-energy TBI cohort had a lower rate of key emergency intervention (LE cohort, 7.5% [95% CI 5.4% to 9.5%], versus HE cohort, 13% [95% CI 12% to 15%]; $p < 0.001$). This was observed for interventions associated with critical care (intracranial pressure

monitoring and decompressive craniectomy) and extracranial injuries (external limb fracture fixation); however, the craniotomy rate was greater in the low-energy TBI cohort (LE cohort, 4.2% [95% CI 2.2% to 6.3%], versus HE cohort, 3.1% [95% CI 1.4% to 4.8%]; $p < 0.001$) (Table 2). In the low-energy TBI cohort, there was a greater time delay between arrival at the study hospital and the provision of emergency interventions such as craniotomy (Table 3); however, the time delay difference was only statically significant for patients receiving craniotomy.

**Outcomes.** Hospital mortality was similar in the high- and low-energy TBI cohorts (6.3% [95% CI 4.2% to 8.3%], versus 7.0% [95% CI 5.3% to 8.6%], respectively; $p = 0.825$); however, there was a greater time to death—from arrival in the study hospital—in low-energy TBI patients than in high-energy TBI patients (median [IQR]: 4 [1.6 to 11] versus 2 [0.7 to 7.5] days; $p < 0.001$). The rate of discharge home was lower (68% [95% CI 67% to 69%] versus 72% [95% CI 72% to 73%]; $p < 0.001$), and to nursing homes higher (9.7% [95% CI 7.7% to 12%] versus 2.2% [95% CI 0.9% to 3.9%]; $p < 0.001$), in the low-energy TBI cohort compared with the high-energy TBI cohort, with similar proportions referred for rehabilitation or transferred to other hospitals (Table 4). When care pathway was accounted for, mortality in the low-energy injury cohort was 6% higher in ICU patients (LE cohort, 22% [95% CI 17% to 28%], versus HE cohort, 16% [95% CI 13% to 20%]) and 4 times greater in admitted patients (LE cohort, 4.2% [95% CI 1.1 to 7.3%], versus HE cohort, 0.9% [95% CI 0.0% to 3.8%]; Fig 3).

The observed differences in rates of critical care admission, and the 4-fold greater in-hospital mortality rate in patients with low-energy TBI admitted to the ward, were unexpected and prompted exploratory assessment of the contribution of energy transfer level to likelihood of critical care/hospital admission and mortality—in risk-adjusted analyses that were not prespecified. The 50% reduction in likelihood of critical care provision for the low-energy TBI cohort persisted after variables influencing critical care admission decisions were adjusted for—adjusted odds ratio 0.46 (95% CI 0.43 to 0.50) to 0.77 (95% CI 0.53 to 1.12) after accounting for the interaction between age and energy transfer mechanism (S3 Table; S3 Fig). The multivariable logistic regression adjusted for demographics (age and sex), injury (ED arrival GCS and intubation status, ED pupillary responses, Marshall classification of CT findings, and AIS grading of extracranial injury severity), and comorbid status (pre-existing health and anticoagulation). A reduction in the likelihood of hospital admission was also observed for patients injured by low- when compared to high-energy transfer, albeit less so in older people (S4 Table; S4 Fig). In patients with TBI admitted to the ward or intensive care, the characteristics more often associated with patients injured by low-energy transfer (older age, pre-injury comorbidity and anticoagulation, and—in ward admissions—having non-evacuated mass lesion [Marshall VI]) were strong independent predictors of hospital mortality; after adjustment for these (and injury severity variables predicting hospital mortality after TBI [18]), low-energy transfer did not independently predict mortality (S5 and S6 Tables; S5 and S6 Figs). Non-evacuated intracranial mass lesions (Marshall VI) were present in 3.1% ($n = 118$) and 1.2% ($n = 53$) of ward admission patients injured by low- and high-energy transfer, respectively ($p < 0.001$).

## Discussion

The CENTER-TBI Registry shows that at least 40% of TBI patients presenting to European and Israeli hospitals are injured by low-energy falls. Our results show that patients injured by low-energy transfer mechanisms (falls) and those injured by high-energy transfer mechanisms (mainly road traffic incidents) are very distinct subpopulations. To our knowledge, this is the first pan-Euro/Israeli study to identify and compare these 2 disease cohorts. Compared to the broader past literature, we observed a greater proportion of older adults ($\geq$65 years old)—

almost 39% of all patients presenting with TBI, as opposed to the 10% to 17% previously reported elsewhere [22,28]. This might be attributable to case ascertainment improvements following recommendations that all older head trauma and/or anticoagulated patients with TBI symptoms receive CT scanning, rather than only to ageing of the population [8,29].

We found that patients in the low-energy TBI cohort, compared with the high-energy TBI cohort, were on average 32 years older, more likely to be female, more than 3 times as likely to be taking pre-injury anticoagulant or platelet aggregate inhibitor medication, and less likely to be classified as moderately or severely injured (based on GCS). Nevertheless, both groups showed clinically similar rates of abnormality on CT scan (29% and 32%), acute hospital admission (58% for high energy, 56% for low energy), and hospital mortality (6.3% and 7.0%). However, the low-energy TBI cohort was 50% less likely to receive critical care (12% versus 24%) or emergency intervention (7.5% versus 13%).

## Low-energy TBI cohort

Our finding that 40% of TBI patients presenting to European hospitals are injured by low-energy falls clearly demonstrates that this low-energy TBI cohort forms an important component of TBI and requires targeted prevention strategies [22,30]. Further, the specific features of this cohort have substantial implications for both clinical care and research. Sixty-six percent of patients injured by low-energy mechanisms are over 64 years of age—a common age cut-off in many clinical trials. Such disenfranchisement of older adults in clinical TBI research is inappropriate—on the contrary, dedicated studies are required [31].

Perhaps most importantly, our results suggest a need to review clinical care pathways and priorities for this group of patients. Our finding that the low-energy TBI cohort was 50% less likely to receive critical care (even after adjustment for age and comorbidity; S3 Table) or emergency intervention highlights an apparent 'non-interventional' approach towards patients injured by low-energy TBI across the continent. This was particularly evident in our post hoc analysis of the drivers of the 4-fold higher mortality rate in low-energy TBI patients admitted to the ward. This analysis showed that reduced rates of intervention (as evidenced by a higher incidence of non-evacuated mass lesions), in addition to age, anticoagulation use, and comorbidity, are explanatory low-energy-TBI-associated features, each independently predicting mortality in ward admissions (S5 Table; Fig 3). The lower critical care provision, and longer times from ED arrival to CT brain scan and emergency interventions, appear to implicitly reflect triage decisions. However, our analyses suggest that low-energy TBI patients do receive timely critical care when their presenting consciousness level is impaired (Table 3; Fig 3). The registry variables did not include 'ceiling of care'; hence, it is uncertain whether the apparent non-intervention strategy for the low-energy TBI cohort reflects therapeutic nihilism approach.

The prevalence of TBI patients taking anticoagulant or/and antiplatelet medication within our study is much higher in the low-energy TBI cohort (44%) than the high-energy TBI cohort (13%), with a greater likelihood of hospitalisation. This reflects the high propensity of these patients with significant comorbidity and frailty to sustain TBI from low-level falls; age-related intracranial changes may also challenge assessment by allowing a higher GCS at presentation compared to younger patients with similar intracranial injury [32]. These findings point to the need for a holistic personalised medicine approach for TBI patients requiring multidisciplinary acute care. Such an approach addresses each patient's pre-existing health issues, the specific brain injury sustained, and their interaction. Overall, there was a higher frequency of intracranial haemorrhage in patients taking antiplatelet medication than in patients taking anticoagulants. These results are in accordance with previous studies [3,32,33], and may possibly be explained by the fact that anticoagulant medication can (and should) be reversed, whilst

antiplatelet medication cannot [34]. These findings indicate the need for imaging guidelines to give as much attention to antiplatelet therapy as to anticoagulation therapy. Our exploratory analyses (S5 and S6 Tables) suggest that specific features of the low-energy TBI cohort—having pre-existing health issues and taking anticoagulation medication—are strong independent predictors of hospital mortality in current-day practice. This signals a need to reassess TBI outcome prediction models, which were mostly developed on older data and did not include these factors and in which, importantly, older patients were underrepresented in the development population [22].

The observed equivalent hospital admission and mortality rates for patients injured by low- and high-energy transfer mechanisms challenge the generalisability of the current paradigm of trauma care systems to prioritise patients injured by high-energy mechanisms [8,9]. For older patients with TBI, the assumption that energy transfer is proportionate to severity of intracranial injury does not appear to be valid. A reappraisal of current injury prevention and clinical management policies is indicated [35].

## Strengths and limitations

The CENTER-TBI Registry study had several strengths: the standardised and robust data collection system, large sample size from specialist neuroscience centres, likely representativeness of the study sample (as illustrated in S2 Fig), participation by a large number of hospitals from 18 countries (17 in Europe and Israel), and inclusion of all TBI severities and age groups. Fig 2 illustrates a considerable low-energy TBI disease burden across countries—the median (IQR) prevalence of low-energy TBI by centre was 36% (24% to 50%) (S1 Fig), suggesting our findings do not result from clustering of low-energy TBI patients in a few large centres or in specific countries. Indeed, as one might expect, the highest-recruiting centres had lower rates of patients injured by low-energy transfer—as study centres with larger catchment populations receive a greater proportion of their patients with TBI through EMS prehospital triage prioritisation of high-energy TBI [7,9,36]. The low-resource-intensive data collection for the registry has enabled the creation of a large dataset that includes patients with dementia and other causes of pre-existing cognitive impairment, characterised by validated energy transfer descriptions. The registry data may therefore be more generalisable to TBI populations across Europe—particularly those admitted to ward settings—than the core study.

Rates of disability beyond discharge were not available in the registry; however, the low rates of discharge home (68% in low-energy TBI and 72% in high-energy TBI) are consistent with the significant rates of post-discharge disability reported in the core study [24]. The registry did not record alcohol ingestion, which may contribute to both energy transfer cohorts (falls in the low-energy TBI cohort, and assaults and road traffic collisions in the high-energy TBI cohort). Our estimates for the study population may be subject to bias as a result of missing data and our purposive sampling strategy. The statistical power arising from the large sample size identified some clinically insignificant differences between the cohorts (proportions with ED pathway, abnormal CT, and intracranial injury) as statistically significant. Five percent of individuals in the registry were excluded because mechanism of injury was unknown; this may reflect the clinical reality of TBI patients being 'found' with impaired consciousness or amnesia and a lack of reliable incident witnesses. As a group, the demographic, injury, and outcome characteristics of those excluded suggest this group contained patients injured by high- and low-energy transfer (S7 and S8 Tables). The reality of injury incidents being unwitnessed—particularly in people falling whilst alone at home—means prehospital staff estimate fall height, resulting in possible misclassification; there is also some variation in classification of the height above ground that is considered low-energy transfer [3,8,9]. The CENTER-TBI

study centres are generally specialist hospitals that receive a high proportion of TBI patients by secondary transfer or directly from the scene of EMS triage, bypassing closer non-specialist hospitals; these patients are generally high-energy TBI patients (Table 2; S1 Fig). Therefore, the proportion of patients with TBI injury by low-energy mechanisms across Europe may be greater than 40%; however, external validity is supported by CENTER-TBI hospital arrival times mirroring those from the Trauma Audit and Research Network (S2 Fig) [18,31]. Our analysis of factors explaining the increase in mortality (S5 and S6 Tables) in low-energy TBI ward and ICU admissions was not prespecified in our analysis plan and should be considered hypothesis-generating, requiring a specific 'appropriate intervention' a priori focus in future research.

## Conclusions

Broad overall descriptions mask the heterogeneity of TBI as a disease. We present the largest standardised and consistently reported description of patients with TBI presenting to hospitals across Europe and Israel to our knowledge, highlighting 2 separate disease cohorts. Clinicians and trauma care systems need to recognise the potential for life-threatening TBI in patients injured by low-energy falls—particularly in alert older patients and those taking anticoagulant or antiplatelet medication. Our findings suggest that within the older cohort, TBI triage based on energy transfer may not inform risk of intracranial injury and hospital mortality. Further studies should test the justification for providing lower rates of critical care and emergency intervention for those injured by low-energy mechanisms. Reduction of the burden and impact of TBI can only be achieved through public health policies and guidelines targeted at the prevention and management of TBI resulting from both high- and low-energy mechanisms.

## Supporting information

**S1 Analysis Plan. The aim of this paper is to describe the demographic, injury, and clinical characteristics of high-energy and low-energy TBI patients who presented to 56 CEN-TER-TBI recruiting hospitals in Europe and Israel, including patients discharged from the emergency room (ER) after imaging.**
(DOCX)

**S1 Fig. Cumulative monthly recruitment across 56 sites and prevalence of low-energy transfer injury mechanism in TBI patients in recruiting sites plotted by estimate precision (1/standard error).**
(TIF)

**S2 Fig. Hospital arrival time by energy transfer level in CENTER-TBI Registry and Trauma Audit and Research Network (TARN).**
(TIF)

**S3 Fig. Receiver operating characteristic (ROC) curve multivariable analysis of factors predicting ICU admission in 11,673[*] patients from the CENTER-TBI Registry.** [*]Excluding 9,286 patients who were discharged from or died in the ED and 722 with missing age/GCS sum score and/or extracranial injury details.
(TIF)

**S4 Fig. Receiver operating characteristic (ROC) curve multivariable analysis of factors predicting hospital admission in 18,035[*] patients from the CENTER-TBI Registry.**
[*]Excluding patients who died in the ED ($n$ = 69), arrived as secondary transfers ($n$ = 2,706), or

had missing age/CT.
(TIF)

**S5 Fig. Receiver operating characteristic (ROC) curve multivariable analysis of factors predicting in-hospital mortality in 7,792**[*] **ward admission patients from the CENTER-TBI Registry.** [*]Excluding 432 patients with missing GCS sum score and/or discharge status.
(TIF)

**S6 Fig. Receiver operating characteristic (ROC) curve multivariable analysis of factors predicting in-hospital mortality in 3,739**[*] **ICU admission patients from the CENTER-TBI Registry.** [*]Excluding 432 patients with missing GCS sum score/age and/or discharge status.
(TIF)

**S1 STROBE Checklist. STROBE statement—Checklist of items that should be included in reports of cohort studies.**
(DOCX)

**S1 Table. Hospitals recruiting to the CENTER-TBI Registry.**
(DOCX)

**S2 Table. Comparison of demographic and comorbid characteristics by energy transfer level and care pathway.** All within-pathway low- versus high-energy differences in age, sex, pre-existing health, and anticoagulant/antiplatelet medication were significant ($p < 0.001$). [*]ED = discharged or died in emergency department. [**]ADM = admitted to hospital but did not receive critical care in study hospital. [***]ICU = admitted to hospital and received critical care in study hospital.
(DOCX)

**S3 Table. Multivariable analysis of factors (age, sex, pre-existing disease status, Marshall CT brain injury classification, CT abnormality, ED GCS and pupillary reactivity, presence of significant extracranial injury, presenting to ED intubated, and causal energy transfer mechanism and its interaction with age) predicting ICU admission in 11,673**[*] **patients from the CENTER-TBI Registry.** [*]Excluding 9,286 patients who were discharged from or died in ED and 722 with missing age/GCS sum score and/or extracranial injury details. AUC = 0.90. [**]AOR = 0.46 (95% CI 0.43 to 0.50) when same model omits age × energy transfer interaction—other variable AORs unchanged. CT, computed tomography; ED, emergency department; GCS, Glasgow Coma Score.
(DOCX)

**S4 Table. Multivariable analysis of factors (age, sex, pre-existing disease status, presence of CT brain abnormality, ED GCS $< 15$, presence of significant extracranial injury, presenting to ED intubated, and causal energy transfer mechanism and its interaction with age) predicting hospital admission in 18,035**[*] **patients from the CENTER-TBI Registry.** [*]Excluding patients who died in the ED ($n = 69$), arrived as secondary transfers ($n = 2,706$), or had missing age/CT abnormality/GCS sum score/extracranial injury ($n = 873$). GCS considered as binary category in accordance with guidance for hospital admission [8]. Area under receiver operating characteristic curve (AUC) = 0.81. AIS, Abbreviated Injury Scale; CT, computed tomography; ED, emergency department; GCS, Glasgow Coma Score.
(DOCX)

**S5 Table. Multivariable analysis of factors (age and sex and their interaction, pre-existing disease status, pre-injury anticoagulation status, Marshall CT brain injury classification, ED GCS and pupillary reactivity, presence of significant extracranial injury, and causal**

energy transfer mechanism and its interaction with age) predicting in-hospital mortality in 7,792* ward admission patients from the CENTER-TBI Registry. *Excluding 432 patients with missing GCS sum score and/or discharge status. Area under receiver operating characteristic curve (AUC) = 0.92. AIS, Abbreviated Injury Scale; CT, computed tomography; ED, emergency department; GCS, Glasgow Coma Score.
(DOCX)

**S6 Table. Multivariable analysis of factors (age and sex and their interaction, pre-existing disease status, pre-injury anticoagulation status, Marshall CT brain injury classification, ED GCS and pupillary reactivity, presence of significant extracranial injury, and causal energy transfer mechanism and its interaction with age) predicting in-hospital mortality in 3,739* ICU admission patients from the CENTER-TBI Registry.** *Excluding 432 patients with missing GCS sum score/age and/or discharge status. Area under receiver operating characteristic curve (AUC) = 0.86. AIS, Abbreviated Injury Scale; CT, computed tomography; ED, emergency department; GCS, Glasgow Coma Score.
(DOCX)

**S7 Table. Demographics, injury mechanism, comorbidity-presenting physiology, and care pathway—Comparative analysis of the CENTER-TBI Registry high-energy, low-energy, and unknown-energy TBI cohorts.** GCS, Glasgow Coma Score. *ED = discharged from or died in the emergency department. **ADM = admitted to hospital but did not receive critical care in study hospital. ***ICU = admitted to hospital and received critical care in study hospital.
(DOCX)

**S8 Table. Imaging findings, injury severity, therapeutic interventions, discharge status—Comparative analysis of the CENTER-TBI Registry high, low and unknown energy transfer cohorts.** AIS, Abbreviated Injury Scale; ASH, acute subdural haematoma; CT, computed tomography; EDH, extradural haematoma; ICP, intracranial pressure; ISS, Injury Severity Score. **Denominator is those with abnormal CT brain scan.
(DOCX)

**S1 Text. Institutional affiliations of CENTER-TBI Participants and Investigators.**
(DOCX)

## Acknowledgments

CENTER-TBI Participants and Investigators (see S1 Text for institutional affiliations 1–145): Cecilia Åkerlund[1], Krisztina Amrein[2], Nada Andelic[3], Lasse Andreassen[4], Audny Anke[5], Anna Antoni[6], Gérard Audibert[7], Philippe Azouvi[8], Maria Luisa Azzolini[9], Ronald Bartels[10], Pál Barzó[11], Romuald Beauvais[12], Ronny Beer[13], Bo-Michael Bellander[14], Antonio Belli[15], Habib Benali[16], Maurizio Berardino[17], Luigi Beretta[9], Morten Blaabjerg[18], Peter Bragge[19], Alexandra Brazinova[20], Vibeke Brinck[21], Joanne Brooker[22], Camilla Brorsson[23], Andras Buki[24], Monika Bullinger[25], Manuel Cabeleira[26], Alessio Caccioppola[27], Emiliana Calappi[27], Maria Rosa Calvi[9], Peter Cameron[28], Guillermo Carbayo Lozano[29], Marco Carbonara[27], Simona Cavallo[17], Giorgio Chevallard[30], Arturo Chieregato[30], Giuseppe Citerio[31,32], Hans Clusmann[33], Mark Coburn[34], Jonathan Coles[35], Jamie D. Cooper[36], Marta Correia[37], Amra Čović[38], Nicola Curry[39], Endre Czeiter[24], Marek Czosnyka[26], Claire Dahyot-Fizelier[40], Paul Dark[41], Helen Dawes[42], Véronique De Keyser[43], Vincent Degos[16], Francesco Della Corte[44], Hugo den Boogert[10], Bart Depreitere[45], Đula Đilvesi[46], Abhishek Dixit[47], Emma Donoghue[22], Jens Dreier[48], Guy-Loup Dulière[49], Ari Ercole[47], Patrick Esser[42], Erzsébet Ezer[50], Martin Fabricius[51], Valery L. Feigin[52], Kelly Foks[53], Shirin Frisvold[54], Alex Furmanov[55], Pablo Gagliardo[56], Damien

Galanaud[16], Dashiell Gantner[28], Guoyi Gao[57], Pradeep George[58], Alexandre Ghuysen[59], Lelde Giga[60], Ben Glocker[61], Jagoš Golubovic[46], Pedro A. Gomez[62], Johannes Gratz[63], Benjamin Gravesteijn[64], Francesca Grossi[44], Russell L. Gruen[65], Deepak Gupta[66], Juanita A. Haagsma[64], Iain Haitsma[67], Raimund Helbok[13], Eirik Helseth[68], Lindsay Horton[69], Jilske Huijben[64], Peter J. Hutchinson[70], Bram Jacobs[71], Stefan Jankowski[72], Mike Jarrett[21], Ji-yao Jiang[58], Faye Johnson[73], Kelly Jones[52], Mladen Karan[46], Angelos G. Kolias[70], Erwin Kompanje[74], Daniel Kondziella[51], Evgenios Kornaropoulos[47], Lars-Owe Koskinen[75], Noémi Kovács[76], Ana Kowark[77], Alfonso Lagares[62], Linda Lanyon[58], Steven Laureys[78], Fiona Lecky[79,80], Didier Ledoux[78], Rolf Lefering[81], Valerie Legrand[82], Aurelie Lejeune[83], Leon Levi[84], Roger Lightfoot[85], Hester Lingsma[64], Andrew I. R. Maas[43], Ana M. Castaño-León[62], Marc Maegele[86], Marek Majdan[20], Alex Manara[87], Geoffrey Manley[88], Costanza Martino[89], Hugues Maréchal[49], Julia Mattern[90], Catherine McMahon[91], Béla Melegh[92], David Menon[47], Tomas Menovsky[43], Ana Mikolic[64], Benoit Misset[78], Visakh Muraleedharan[58], Lynnette Murray[28], Ancuta Negru[93], David Nelson[1], Virginia Newcombe[47], Daan Nieboer[64], József Nyirádi[2], Otesile Olubukola[79], Matej Oresic[94], Fabrizio Ortolano[27], Aarno Palotie[95,96,97], Paul M. Parizel[98], Jean-François Payen[99], Natascha Perera[12], Vincent Perlbarg[16], Paolo Persona[100], Wilco Peul[101], Anna Piippo-Karjalainen[102], Matti Pirinen[95], Horia Ples[93], Suzanne Polinder[64], Inigo Pomposo[29], Jussi P. Posti[103], Louis Puybasset[104], Andreea Radoi[105], Arminas Ragauskas[106], Rahul Raj[102], Malinka Rambadagalla[107], Jonathan Rhodes[108], Sylvia Richardson[109], Sophie Richter[47], Samuli Ripatti[95], Saulius Rocka[106], Cecilie Roe[110], Olav Roise[111,112], Jonathan Rosand[113], Jeffrey V. Rosenfeld[114], Christina Rosenlund[115], Guy Rosenthal[55], Rolf Rossaint[77], Sandra Rossi[100], Daniel Rueckert[61] Martin Rusnák[116], Juan Sahuquillo[105], Oliver Sakowitz[90,117], Renan Sanchez-Porras[117], Janos Sandor[118], Nadine Schäfer[81], Silke Schmidt[119], Herbert Schoechl[120], Guus Schoonman[121], Rico Frederik Schou[122], Elisabeth Schwendenwein[6], Charlie Sewalt[64], Toril Skandsen[123,124], Peter Smielewski[26], Abayomi Sorinola[125], Emmanuel Stamatakis[47], Simon Stanworth[39], Robert Stevens[126], William Stewart[127], Ewout W. Steyerberg[64,128], Nino Stocchetti[129], Nina Sundström[130], Riikka Takala[131], Viktória Tamás[125], Tomas Tamosuitis[132], Mark Steven Taylor[20], Braden Te Ao[52], Olli Tenovuo[103], Alice Theadom[52], Matt Thomas[87], Dick Tibboel[133], Marjolein Timmers[74], Christos Tolias[134], Tony Trapani[28], Cristina Maria Tudora[93], Andreas Unterberg[90], Peter Vajkoczy[135], Shirley Vallance[28], Egils Valeinis[60], Zoltán Vámos[50], Mathieu van der Jagt[136], Gregory Van der Steen[43], Joukje van der Naalt[71], Jeroen T. J. M. van Dijck[101], Thomas A. van Essen[101], Wim Van Hecke[137], Caroline van Heugten[138], Dominique Van Praag[139], Thijs Vande Vyvere[137], Roel P. J. van Wijk[101], Alessia Vargiolu[32], Emmanuel Vega[83], Kimberley Velt[64], Jan Verheyden[137], Paul M. Vespa[140], Anne Vik[123,141], Rimantas Vilcinis[132], Victor Volovici[67], Nicole von Steinbüchel[38], Daphne Voormolen[64], Petar Vulekovic[46], Kevin K. W. Wang[142], Eveline Wiegers[64], Guy Williams[47], Lindsay Wilson[69], Stefan Winzeck[47], Stefan Wolf[143], Zhihui Yang[113], Peter Ylén[144], Alexander Younsi[90], Frederick A. Zeiler[47,145], Veronika Zelinkova[20], Agate Ziverte[60], Tommaso Zoerle[27].

We gratefully acknowledge interactions and support from International Initiative for Traumatic Brain Injury Research funders and investigators. We are immensely grateful to our patients with TBI for helping us in our efforts to improve care and outcomes for TBI. We also acknowledge the contribution and expert advice of Dr Richard Jacques and Joanne Palfreman (University of Sheffield, UK) in conducting statistical analyses (RJ) and manuscript preparation (JP).

## Role of Sponsor

Antwerp University Hospital is the study co-ordination centre and undertook the role of sponsor, ensuring research governance according to international standards across the 56

recruiting centres in study procedures, specifically for enrolling patients into the CENTER-TBI Registry and the storage and analysis of patient data.

## Transparency statement

Fiona E. Lecky and Andrew I. R. Maas (the paper's guarantors) affirm that the paper is an honest, accurate, and transparent account of the study being reported; that no important aspects of the study have been omitted; and that there are no discrepancies from the study as originally planned.

## Author Contributions

**Conceptualization:** Fiona E. Lecky, Olubukola Otesile, Carl Marincowitz, Andrew I. R. Maas.

**Data curation:** Fiona E. Lecky, Olubukola Otesile, Andrew I. R. Maas.

**Formal analysis:** Olubukola Otesile, Carl Marincowitz.

**Funding acquisition:** Fiona E. Lecky, Marek Majdan, Daan Nieboer, Hester F. Lingsma, Marc Maegele, Giuseppe Citerio, Nino Stocchetti, Ewout W. Steyerberg, David K. Menon, Andrew I. R. Maas.

**Investigation:** Fiona E. Lecky, Marek Majdan, Daan Nieboer, Hester F. Lingsma, Marc Maegele, Giuseppe Citerio, Nino Stocchetti, Ewout W. Steyerberg, David K. Menon, Andrew I. R. Maas.

**Methodology:** Fiona E. Lecky, Olubukola Otesile, Carl Marincowitz, Marek Majdan, Daan Nieboer, Hester F. Lingsma, Marc Maegele, Giuseppe Citerio, Nino Stocchetti, Ewout W. Steyerberg, David K. Menon, Andrew I. R. Maas.

**Project administration:** Fiona E. Lecky, Olubukola Otesile, Marek Majdan, Daan Nieboer, Hester F. Lingsma, Marc Maegele, Giuseppe Citerio, Nino Stocchetti, Ewout W. Steyerberg, David K. Menon, Andrew I. R. Maas.

**Resources:** Fiona E. Lecky, Olubukola Otesile, Marek Majdan, Daan Nieboer, Hester F. Lingsma, Marc Maegele, Giuseppe Citerio, Nino Stocchetti, Ewout W. Steyerberg, David K. Menon.

**Software:** Hester F. Lingsma, Ewout W. Steyerberg.

**Supervision:** Fiona E. Lecky, Andrew I. R. Maas.

**Validation:** Fiona E. Lecky, Olubukola Otesile, Carl Marincowitz.

**Visualization:** Olubukola Otesile.

**Writing – original draft:** Fiona E. Lecky, Olubukola Otesile, Andrew I. R. Maas.

**Writing – review & editing:** Fiona E. Lecky, Olubukola Otesile, Carl Marincowitz, Marek Majdan, Daan Nieboer, Hester F. Lingsma, Marc Maegele, Giuseppe Citerio, Nino Stocchetti, Ewout W. Steyerberg, David K. Menon, Andrew I. R. Maas.

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
