## [Editor Report · Decision Letter 0]

27 Oct 2020

Dear Dr Lecky, 

Thank you for submitting your manuscript entitled "Traumatic Brain Injury Disease Burden from low energy falls: Findings of the CENTER TBI Registry" for consideration by PLOS Medicine.

Your manuscript has now been evaluated by the PLOS Medicine editorial staff as well as by an academic editor with relevant expertise, and I am writing to let you know that we would like to send your submission out for external peer review.

Kind regards,

Caitlin Moyer, Ph.D.,

Associate Editor

PLOS Medicine

---

## [Decision Letter · Decision Letter 1]

9 Apr 2021

Dear Dr. Lecky,

Thank you very much for submitting your manuscript "Traumatic Brain Injury Disease Burden from low energy falls: Findings of the CENTER TBI Registry" (PMEDICINE-D-20-05160R1) for consideration at PLOS Medicine. 

Your paper was evaluated by a senior editor and discussed among all the editors here. It was also discussed with an academic editor with relevant expertise, and sent to three independent reviewers, including a statistical reviewer. The reviews are appended at the bottom of this email and any accompanying reviewer attachments can be seen via the link below:

[LINK]

In light of these reviews, I am afraid that we will not be able to accept the manuscript for publication in the journal in its current form, but we would like to consider a revised version that addresses the reviewers' and editors' comments. Obviously we cannot make any decision about publication until we have seen the revised manuscript and your response, and we plan to seek re-review by one or more of the reviewers. 

We expect to receive your revised manuscript by Apr 30 2021 11:59PM. Please email us (plosmedicine@plos.org) if you have any questions or concerns.

We look forward to receiving your revised manuscript. 

Sincerely,

Caitlin Moyer, Ph.D.

Associate Editor 

PLOS Medicine

plosmedicine.org

1. Competing Interests: Please add this statement to the manuscript's Competing Interests: "DM is an Academic Editor on PLOS Medicine's editorial board."

2. Data availability statement: Please revise the statement. a) If the data are freely or publicly available, note this and state the location of the data: within the paper, in Supporting Information files, or in a public repository (include the DOI or accession number).

We suggest revising the note regarding data access as: “Data cannot be shared due to XXX. Proposals to access the study data, data dictionary, analytic code, and analysis scripts may be submitted online at https://www.center-tbi.eu/data. Proposals are subject to review by the management committee. A Data Access Agreement is required, and all access must comply with regulatory restrictions imposed on the original study.” or similar.

3. Abstract: Please structure your abstract using the PLOS Medicine headings (Background, Methods and Findings, Conclusions).

4. Abstract: Please quantify the main results (with 95% CIs and p values). In the last sentence of the Abstract Methods and Findings section, please describe the main limitation(s) of the study's methodology.

5. Abstract: Conclusions: Please address the study implications without overreaching what can be concluded from the data; beginning the section with the phrase "In this study, we observed ..." may be useful.

6. Author Summary: Thank you for including the summary box. Please reformat this into a short, bulleted, non-technical Author Summary of your research to make findings accessible to a wide audience that includes both scientists and non-scientists. The Author Summary should immediately follow the Abstract in your revised manuscript. This text is subject to editorial change and should be distinct from the scientific abstract. Please see our author guidelines for more information on Author Summary format: https://journals.plos.org/plosmedicine/s/revising-your-manuscript#loc-author-summary

7. References: Please use square brackets for in-text citations, like this [1]. Please do not include spaces within brackets when listing multiple references [1-4].

8. Line numbers: Please include line numbers throughout text when re-submitting the document.

9. Methods: Page 7: Please include in the text, the 17 countries included. If possible, please include the information/distributions for the 56 participating centres as supporting information.

10. Methods: Page 8: It would be helpful to have more detail on the nature of the data collected: “We collected variables describing demographics (age & gender), pre-existing health status (https://www.asahq.org/standards-and-guidelines/asa-physical-status-classification-system), mechanism of injury, injury severity descriptors (Glasgow Coma Scale (GCS), Abbreviated Injury Scale (AIS)), presenting physiological vital signs, CT Brain findings, processes of care, and immediate outcome of care in terms of hospital mortality and destination on discharge (Tables 1-4). These variables were derived from the Utstein trauma template used for standard trauma registry collection across Europe, North America and Australasia.”

11. Methods: Page 9: Please note how you accounted for clustering at the country level (and center within country).

12. Methods: Page 9: Please explicitly indicate which analyses were done using chi-square, and which were done using Mann-Whitney tests, at least in the Table legends where the results are presented. And please clarify if all the continuous variables were non-parametric.

13. Methods: Did your study have a prospective protocol or analysis plan? Please state this (either way) early in the Methods section.

14. Methods: Page 10: Please revise the STROBE statement to: "This study is reported as per the Strengthening the Reporting of Observational Studies in Epidemiology (STROBE) guideline (S1 Checklist)."

15. Results: Page 11: Please define the abbreviations “PM” and “TARN” at first use.

16. Results: Page 11: Please clarify if the “Overall Cohort” results described and Table 1 pertain to the overall registry, or specifically those included in the study.

17. Results: Please provide 95% CIs and p values for all analyses described in the text.

18. Results: Page 18: The multivariable logistic regression here did not seem to be described in the Methods section. For the following result, please indicate the variables used in the adjustment: “The 50% reduction in likelihood of critical care provision for the low energy cohort persisted after demographic, injury and comorbid considerations were accounted for in multivariable logistic regression - Adjusted Odds Ratio (AOR) 0·46 (95% Confidence Interval (CI) (0·43-0·50), a reduction in the likelihood of hospital admission was also observed for patients injured by low energy transfer albeit less so in older people. In patients with TBI admitted to the ward and intensive care the characteristics of patients injured by low energy transfer ( older age, pre-injury comorbidity and anticoagulation, non-evacuated mass lesion – Marshall VI- in ward admissions) were strong independent predictors of hospital mortality; after adjustment for these low energy transfer did not independently predict mortality (suppl tables iv-vii); non-evacuated intracranial mass lesions were present in 3·1% (n=118) and 1·2% (n=53) of ward admission patients injured by low and high energy transfer.”

19. References: Please use the "Vancouver" style for reference formatting, and see our website for other reference guidelines https://journals.plos.org/plosmedicine/s/submission-guidelines#loc-references

20. Table 1: Please define all abbreviations in the legend (TBI, IQR).

21. Supporting information files: Please include both a descriptive title and legend for each figure/table of the supporting information. In particular, please provide a description for Supporting Information Figure S2.

22. Supporting information Tables S4, S5, S6, S7: In the legend please provide the variables used in the adjustment. Please also present the unadjusted results.

23. STROBE Checklist: Please reformat the checklist, using section and paragraph numbers to refer to locations within the text, rather than page numbers.

Comments from the reviewers:

Reviewer #1: The authors presented a well-constructed study addressing TBI Disease Burden focusing in low energy falls as mechanism of injury using data from the CENTER TBI registry. Whilst the data are interesting, they are very similar to those shown by a huge amount of studies evaluating geriatric TBI (actually, those patients who undergone low energy falls). 

Since the study underscores that patients with low energy falls receive less invasive monitorization/treatments, it would be interesting to know how many of these patients had any limitation of life sustaining therapies. This would help to better characterize the population. 

Minor comments: 

Some references must be updated.

References are presented in different formats

Reviewer #2: "Traumatic Brain Injury Disease Burden from low energy falls: Findings of the CENTER TBI Registry" describes a comparative cohort study on the CENTER-TBI registry, involving over 21,000 patients presenting with traumatic brain injury (TBI) and indications for CT brain scan enrolled from 18 countries, from 2014 to 2018. The effect of various patient characteristics (both unadjusted and adjusted) is examined, for three clinical care pathways - emergency department (ED), admission to hospital but not ICU (ADM), and admission to the ICU (ICU). The main conclusion was that patients sustaining TBI comprise an important distinct demographic cohort, and that the level of energy transfer should not inform triage since it does not predict intracranial injury severity or threat to life.

While the scale of the registry and study is impressive, there are a number of issues that might be addressed:

1. Given the main recommendation of the study ("level of energy transfer should not inform triage"), it might be clarified as to the degree (if any) that level of energy transfer currently informs triage in the participating countries/hospital, cited guidelines ([8],[9]) notwithstanding.

In particular, while Table 1 affirms that level of energy transfer does not strongly predict intracranial injury severity (at least as measured by GCS), it can be observed that the care pathway distribution differs significantly between high and low energy TBI (24.4% ICU for high, versus 11.5% ICU for low). However, it is not certain whether this difference in clinical pathway is due to triaging on level of energy transfer in the first place, or due to other criteria. Indeed, it is mentioned that "most published TBI series do not differentiate high from low energy transfer falls" (Page 6). This might be clarified.

2. Additionally, one might suspect that injury events that are classified as high energy transfer (e.g. traffic collisions, falling from a height, etc) may be more likely to be triaged as ICU, at least partly due to accompanying critical injury to other organs/limbs (e.g. internal injury to core organs, broken arms/legs). It is unclear whether these possible non-brain/head injuries were record/accounted for in adjustments, from the patient characteristics presented in Table 1, despite possibly being a major factor in triage outcomes in practice. This might be clarified.

3. On Page 7, the Inclusion/Exclusion criteria might be made more consistent with Figure 1. In particular, it is stated that "...the Registry included patients with pre-existing cognitive impairment etc... these three groups were exclused from recruitment to the CENTER TBI core study", while Figure 1 appears to describe the bulk of the exclusions (1101 patients) as "missing and unknown mechanisms of injury". Terminology might be standardized between the main text and the flowchart figure, and it might be made more explicit in the figure as to what the registry includes (the initial 22,849 patients?) and where the recruitment to the core study is complete (the 21,681 patients?)

4. More generally, the existing triage criteria for assigning a patient to one of the three clinical pathways (ED, ADM, ICU; Page 8) might be explained in greater detail (partly alluded to in Point 1). If clinicians considered level of energy transfer as a criteria for triage, was it a major or minor criteria? 

5. The purposive sampling procedure (Page 7) might be described in greater detail. How was it designed to produce a representative sample?

6. On Page 8, the ED clinical pathway is described as "either discharge or direct to the mortuary from the ED"; it might be clarified as to whether "discharge from the ED" in this case, is a full discharge from the hospital, or possibly to general admissions/the ICU.

7. More details on the multivariate analyses (as presented in Supplementary Tables V to VII) might be provided. In particular, how were the variables involved for each of these analyses selected, given that some available variables from Tables 1 & 2 appear to be omitted? p-values might also be included.

8. For EDGCS in Supplementary Tables V to VII, it appears presented as a binary category despite involving multiple groups/classes (as described in Table 1; contrast the Marshall Classification groups, which are analyzed individually). As such, the delineation of these groups/classes into a binary categorization might be described and justified.

9. Still on Supplementary Tables V to VII, an AUC value (e.g. AUC = 0.90) is mentioned for each table in its caption. If these AUCs pertain to each assumed logistic regression model in predicting ICU/mortality, it might be considered to also present the corresponding ROC curves.

10. A general reservation is that while the manuscript suggests that TBI triage on energy transfer does not inform risks of intracranial injury/hospital mortality and thus should be discarded, it does not appear to propose and justify a concrete alternative. In particular, the presented multivariate analyses would seem to have the potential to quantitatively suggest an alternative triage model (or risk score, since it is admitted in the limitations section that the analyses are post-hoc; therefore, the impact of improved triage is probably difficult to estimate). This might be considered by the authors.

11. There are a number of phrasing issues, e.g. the sentences "Injury classification by energy transfer mechanisms..."; "Patients injured by high energy mechanisms being conveyed..." (Page 5), "Forty percent (8622) of patients..." (Page 15), "significant longer term rates of disability..." (Page 22), etc.

Reviewer #3: Markus Skrifvars, University of Helsinki

Regarding the paper "Traumatic Brain Injury Disease Burden from low energy falls: Findings of the CENTER TBI Registry" submitted to PLOS Medicine. This a sub-study of the large TBI registry CENTER-TBI. This paper contains data from a large database/study with potential interest to the general medical audience. I have the following comments: 

- The major aim of this study is to compare low and high energy trauma. Therefore it is paramount to include more precise information on how this classification was done. Perhaps include a Table with the different types of in the two groups? Was this classification done on site? Was there any data validation between centers in order to make sure that different centers scored this the same way? Of course one may debate whether a low velocity car accident or bicycle accident (with a helmet) is different from a fall from standing were the energy on the head may be substantial. 

- The manuscript mentions that patients with a suspected or confirmed TBI who had a CT brain done were included. Did there need to be a clear presented history of a trauma? I refer to the elderly patient who is found at home without any clear evidence of what has happened. Continuing on from this do you have information on how many of the abnormal CT brain findings were unexpected? Again many of these patients will have an unclear event history with many factors making the diagnostic work up more difficult (intoxiocations, seazures, cardiac arrhythmias etc.).

- One would intuitively think that the use of "not for ICU", DNR and other treatment limitations would impact the admission of the elderly patients with multiple comorbidities to ICU. Is this information included in the registry?

- I would personally like to see the results of a multivariable model predicting outcome including high/low energy. This could even be done for only the patients with an abnormal CT scan. The key question is, if you look at two similar abnormal CT scans will the outcome be different if they are the result of a high or low energy? When you have the abnormal CT in front of you what are the factors that drive admission to ICU? I suspect it is age. 

- Continuing on from the previous point. I would be interested to know if one ends up in the ICU does it then make any difference for outcome (survival, neurological outcome) if the energy was high or low?

- Do you have information about frailty and functional capacity? Again I suspect that this piece of information will drive the decision on further care. 

- In the first sentence of the discussion there is a statement that 40% of patients with TBI have a low energy injury pattern. Does this comparison also include those with a negative CT brain? Perhaps state more specifically about numbers in those with a CT brain confirmed TBI?

- You mention in the discussion that the low-energy patients may be receive less intense care. This may be the case but is this the results of the injury being of low energy or the fact that these patients are not perceived to benefit from these interventions? Unless you present data that many of these "low energy" patients experience late deterioration with emergency surgery and ICU admission this may not be completely supported by your data. 

- The European map with differences in incidence of low energy TBI is intriguing. However, this is not discussed in the manuscript. Were the centers included in the CENTER-TBI study of comparable enabling the drawing of such a Figure? Indeed one would like to see the equivalent of a 95% confidence interval in the colors. This is of course not possible but somehow certainly of these findings must be presented. 

- Do you have information on how many patients were intoxicated? Again data suggests the association between alcohol/drugs and TBI from low-energy trauma. 

- In the discussion the authors state that anti-platelet medication effects cannot be reversed. How the administration of platelets? 

- The conclusion includes some ideas about the need for future studies especially regarding the need for more studies on the intensity of TBI care in low-energy cohort. Do we really think that it is the injury energy in itself that is driving care decisions and not the patient? I suspect that most clinicians make their decision on a combination of things with patient age, comorbidities and functional status being in focus. So unless the authors have compelling data that really challenges this, perhaps the conclusion should be softened.

[LINK]

---

## [Decision Letter · Decision Letter 2]

15 Jun 2021

Dear Dr. Lecky,

Thank you very much for submitting your revised manuscript "Traumatic Brain Injury Disease Burden from low energy falls: Findings of the CENTER TBI Registry" (PMEDICINE-D-20-05160R2) for consideration at PLOS Medicine. 

Your paper was evaluated by a senior editor and discussed among all the editors here. It was also discussed with an academic editor with relevant expertise, and sent to two of the original reviewers, including a statistical reviewer. The reviews are appended at the bottom of this email and any accompanying reviewer attachments can be seen via the link below:

[LINK]

In light of the reviewer comments, we will not be able to accept the manuscript for publication in the journal in its current form, but we would like to consider a revised version that addresses the reviewers' and editors' comments. Obviously we cannot make any decision about publication until we have seen the revised manuscript and your response, and we plan to seek re-review by one or more of the reviewers. 

We expect to receive your revised manuscript by Jun 29 2021 11:59PM. Please email us (plosmedicine@plos.org) if you have any questions or concerns.

We look forward to receiving your revised manuscript. 

Sincerely,

Caitlin Moyer, Ph.D.

Associate Editor 

PLOS Medicine

plosmedicine.org

1. Please completely address the points of the reviewers.

2. Data availability statement: Please remove the initial “Data sharing statement:” from this section.

3. Title: Please revise your title according to PLOS Medicine's style. Your title must be nondeclarative and not a question. It should begin with main concept if possible. "Effect of" should be used only if causality can be inferred, i.e., for an RCT. Please place the study design ("A randomized controlled trial," "A retrospective study," "A modelling study," etc.) in the subtitle (ie, after a colon).

We suggest: “Traumatic brain injury disease burden from low energy falls among patients from 17 countries from the CENTER-TBI Registry: A comparative cohort study” or similar, mentioning the design and setting.

4. Abstract: Please combine the Methods and Findings into a single section, with the main limitations as the final sentence of this section.

5. Abstract: Line 61-62: Please reword this sentence if possible, to clarify.

6. Abstract: Lines 65-76: Where the findings are reported, it is not completely clear that the first group is the low energy and the second is the high energy cohort. Please clarify this if possible.

7. Abstract: Line 71-72: Please clarify whether the finding of CT abnormalities was or was not significantly different between groups (p<0.001 is reported, where the text reports the groups were similar).

8. Abstract: Line 81-82: We suggest tempering this conclusion, given some of the limitations pointed out by the reviewer. “This suggests energy transfer level should no longer inform prehospital and emergency department TBI triage in older people.”

9. Author Summary: Please remove the “summary box” and please format using bullet points (we suggest 3-4 for each of the three sections). The third section should be titled “What do these findings mean?”

10. Author Summary: Please temper the second and third points of the “What do these findings mean” section, please interpret the study based on the results presented in the abstract, without overstating your conclusions.

11. Methods: Analysis Plan: Thank you for describing the analysis plan for your study. If feasible, please include a document version of the analysis plan as a supporting information file.

12. Results: For the “Comparisons by Energy Transfer mechanism” and “Injury characteristics” sections, please make it more clear in the text where possible, the nature of the two groups being compared (for example, where two sets of values refer to the low and high energy transfer groups, respectively).

13. Results: Line 314-317: Please clarify if these differences reached statistical significance (Table 2 seems to suggest they may). “In patients injured by low energy falls the proportion with abnormalities detected on CT scan (including skull fracture) and proportion of abnormal scans showing intracranial injury (29% (95% CI 27-31%) and 70% (95% CI 69-73%) was similar to that of patients injured by high energy mechanisms (32% (95% CI 31-34%) and 75% (95% CI 74-77%)).

14. Results: Line 342-344: Please clarify that for this finding, this was significant only for specific intervention (craniotomy) as shown in Table 3. “In the low energy cohort there was a greater time delay between arrival at the study hospital and the provision of emergency interventions such as craniotomy (Table 3).”

15. Results: Line 371-378: Please revise this sentence to clarify, or split into multiple sentences.

16. Discussion: Line 406: Please clarify this sentence to reflect the study findings. In your analysis, there seemed to be a significant differences in CT abnormality between the two groups, for example. You note later in the limitations section that these may not be clinically meaningful, but perhaps that could be expanded on/clarified here.

17. Conclusions: Line 511: Please modify this sentence with “...to our knowledge, the largest…” or similar wording.

18. Conclusions: Line 515: Please revise this sentence to temper the recommendations somewhat: Our findings suggest that within the older cohort, TBI triage based on energy transfer may not inform risk of intracranial injury and hospital mortality.”

19. Acknowledgements: Please remove the funding information from this section, and ensure all information is accurately entered into the Financial Disclosure and Competing Interests section of the manuscript submission form.

20. Page 28-29: Please remove the “Financial Disclosure” and “Competing Interests” sections from the main text of the manuscript- this information should be accurately entered into the manuscript submission form.

21. Line 583: List of The CENTER-TBI participants and investigators. Please include those who contributed to the work but do not meet our authorship criteria should be listed in the Acknowledgements with a description of the contribution. Authors are responsible for ensuring that anyone named in the Acknowledgements agrees to be named.

22. Figure 1: Please include a descriptive legend, including a definition of RTC.

23. Figure 2: Please confirm that the appropriate usage rights apply to the use of this map. Please see our guidelines for map images: https://journals.plos.org/plosmedicine/s/figures#loc-maps

24. Figure 3: Please include a legend, including definitions for all abbreviations used.

25. Table 2: Please clarify ASH vs ASDH in the table/legend.

26. Checklist: Please remove all references to page numbers.

Comments from the reviewers:

Reviewer #2: We thank the authors for addressing the previously raised points, and concerns over potential confounding on severity of extracranial injuries with reference to Supplemental Table V, which appears also updated with univariate odds from the previous revision (as is Supplemental Table IV).

1. It might however be explained as to why some values in the (updated) Supplemental Table V, appear different from the previous revision. For example, adjusted odds ratio for Significant Extracranial Injury is now 1.60 (1.40-1.82), while previously it was 1.59 (1.40-1.81). It might also be considered to explicitly state the baseline reference and condition being varied, for these tables (e.g. baseline being low energy, versus high energy)

2. The revised Results section states that "The 50% reduction in likelihood of critical care provision for the low energy cohort persisted after variables influencing critical care admission decisions were adjusted... Adjusted Odds Ratio (AOR) 0·46 (95% CI 0·43-0·50) - AOR 0.77 ( 95% CI 0.53-1.12) after accounting for the interaction between age and energy transfer mechanism (supplemental table v)". However, it is not clear where the initial 0.46 OR arises from, in the actual Supplemental Table V. The relevant univariate OR appears to be given as 0.36. This might be clarified.

3. The additional ROC curves provided as requested are appreciated. However, Supplemental Figure IV shows an AUROC = 0.9012, while its corresponding Supplemental Table V gives AUC = 0.91, which does not appear to correspond after rounding, unlike the other three figure/table pairs. This might be clarified.

4. It has been partly clarified that "Decisions on the care pathway for patients after the Emergency Department CT scan (decisions on discharge from hospital, or level of care for admitted patients) depend on imaging findings as well as the initial triage priority". Further specific details on the care pathways might be described, if possible.

5. The explanation for Point 8 on EDGCS as a binary category, might also be included in the manuscript or supplementary material in some capacity.

Reviewer #3: Markus Skrifvars University of Helsinki

Regarding the revised version of the paper "Traumatic Brain Injury Disease Burden from low 2 energy falls: Findings of the CENTER TBI Registry" submitted to Plos Medicine. The authors have revised their manuscript. The paper contains important information about the epidemiology if TBI care. However some of the conclusions made and recommendations are not completely supported by the data and further softening is in my mind required. From a societal/research perspective the key issue is the need to include these patients in TBI trials. My personal clinical take from this teh very risky patient group of; the elderly patient using anticoagulants/antiplatelet agents with low energy fall. 

- In some parts of the manuscript I feel that there still is a suggestion that "low energy" in itself is the main reason why the patient does not receive ICU care For example on line 408-409 the authors state "However, the low energy cohort was 50% less likely to receive critical care (12% versus 24%) or emergency intervention (7·5% versus 13%)". I would suggest that one could also use the word "need" instead of "receive". I do not think that a study such as this can provide solid evidence that it is the injury mechanism in itself that prompted the neurosurgeon/neurointensivist NOT to admit the patient. 

- Continuing from the previous comment, what do the authors suggest we should do in case of a low energy TBI patient with a GCS 14-15 patient and with a non-operatively managed injury on the CT scan? What is the ICU intervention that will make a difference? 

- As the authors are comparing high and low energy falls it is quite clear that the reason the high energy group may have needed ICU care not due to their TBI but to other injuries. For example, how many were sedated/mechanically ventilated in the ED? It is quite clear that such a patient will go to the ICU no matter what?

- The statement in conclusion "This suggests energy transfer level should no longer inform prehospital and emergency department TBI triage in older people". This is a bit problematic I think. I think one can say based on these findings that a low energy level does not rule out significant TBI with high morbidity and mortality. But on the other hand, I do think that a significant injury mechanism must still be seen as a major contributor to morbidity and mortality and should treated as such. There is also the risk of multi-trauma. I would maintain that from the pre-hospital standpoint an elderly patient with a low energy fall is more likely to have a mild TBI/other pathology than one injured for example in a high speed traffic collision. 

- It would be very interesting to the see the IMPACT risk in the high and low energy cohorts. In a simple model with the IMPACT risk included is the trauma energy a significant predictor of outcome? Is this the same in different age categories and based on use of antiplatelet agents. 

- The authors mentions alcohol and drug intoxication as a challenge for pre-hospital TBI care. Can the authors perhaps provide a reference for this and slightly expand what they mean? Are intoxicated patients thought to have a more severe TBI or vice versa?

[LINK]

---

## [Decision Letter · Decision Letter 3]

28 Jul 2021

Dear Dr. Lecky,

Thank you very much for re-submitting your manuscript "Traumatic Brain Injury Disease Burden from low energy falls:A comparative cohort study of patients from 18 countries   - the CENTER TBI Registry." (PMEDICINE-D-20-05160R3) for review by PLOS Medicine.

I have discussed the paper with my colleagues and the academic editor and it was also seen again by two reviewers. I am pleased to say that provided the remaining editorial and production issues are dealt with we are planning to accept the paper for publication in the journal.

[LINK]

We look forward to receiving the revised manuscript by Aug 04 2021 11:59PM.   

Sincerely,

Caitlin Moyer, Ph.D.

Associate Editor 

PLOS Medicine

plosmedicine.org

Requests from Editors:

1. Title: We suggest revising the title to: The burden of traumatic brain injury from low energy falls among patients from 18 countries in the CENTER TBI registry: A comparative cohort study”

2. Data availability statement: Please remove the opening set of quotation marks (“Data cannot be shared…).

3. Abstract: Lines 59-60: We suggest reformatting this sentence to clarify, and incorporating it into the following paragraph: “22782 eligible patients were enrolled, and study outcomes were analysed on 21681 TBI patients with known injury mechanism.”

4. Abstract: Line 72-73: Please clarify this sentence to make it clear it is the low energy group who were more likely to not receive critical care and emergency intervention.

5. Abstract: Line 79-81: We suggest revising to “...and further, our findings suggest that energy transfer level may not predict intracranial injury or acute care mortality in patients with TBI presenting to hospital. This suggests factors beyond energy transfer level may be more relevant to prehospital and emergency department TBI triage in older people.” or similar, depending on your meaning.

6. Author summary: Line 91: We suggest revising “are given lower priority” to “can be given lower priority” as it seems like this wouldn’t universally be the case depending on the circumstances.

7. Author summary: Line 101: We suggest revising to: “We found that the 40% of patients with TBI who were injured through low energy falls were significantly older…”

8. Author summary: Line 109-110: We suggest revising to “Low energy falls contribute to a significant portion of the TBI disease burden, which will increase…”

9. Author summary: Line 113-114: We suggest clarifying the wording here that the findings don’t necessarily indicate that fall energy level is not relevant to the triage process.

10. Introduction: Line 124-126: Please clarify to indicate that this is not an all-inclusive list of the sources of high and low energy transfer mechanisms “(High energy transfer resulting from road traffic collisions, falling from a height, blunt assault or contact sports while low energy transfer results from low level falls or those from a standing height [6,7,8])”

11. Introduction: Line 127-128: We suggest revising to: “informing Emergency Medical Service (EMS) on scene trauma triage [3,8,9], with patients injured by high energy transfer…”

12. Methods: Line 208: Please clarify if the energy mechanism classification list here is inclusive of all types of injuries presented.

13. Results: Line 334: Please fix the “s” typo in this sentence: “...was s lower than that of patients”

14. Results: Line 352: Please check if an end parenthesis is missing from: “(median (IQR) extracranial ISS; LE = 2 (0-8) versus HE = 5 (1-13) p<0.001.”

15. Results: Line 381-384: Please clarify the numbers associated with the high energy group in this sentence: “...6% higher in ICU patients (22% (95% CI 17-28%) versus 16 % (95% CI 13-20%)) 383 and four times greater in admitted patients (4.2% (95% CI 1.1–7.3%) versus 0·9% (95% CI 0.0 -3.8%)...”

16. Discussion: Line 407-408 and at Line 425: We suggest making the distinction between European hospitals and hospitals located in Israel. Please similarly adjust the language at line 410-411- “the first pan European study” as it is not clear if Israel is being considered as a European country.

17. Discussion: Line 411-413: Please revise this sentence to clarify if you mean that there are a greater proportion of older adults in the low energy fall cohort or among all of those presenting with TBI: “Compared to the broader past literature we observed a greater proportion of older adults (>65 years old)-almost 39% as opposed to 10-17% previously reported elsewhere [22,27].”

18. Discussion: Line 457: Please remove the superscript formatting applied to the word “and” in the sentence.

19. Discussion: Line 476: Please clarify, here and throughout, if there are 18 or 17 countries represented in the study.

20. Conclusions: Line 515-517 Please also change to “across Europe and Israel” or similar.

21. Page 33: Please remove the “Role of the funding source” section from the main manuscript, and ensure the information is accurately entered into the Financial Disclosures section of the manuscript submission system.

22. Page 33: Please remove the “Data sharing” section from the main manuscript, and ensure the information is accurately entered into the Data Availability section of the manuscript submission system.

23. Acknowledgements: We suggest listing all contributing members of the group, with affiliations, in a supporting information file. Or, including the names in the acknowledgements, but moving the affiliations to a supporting information file.

24. Figure 1: Please remove the extra paragraph symbols from the figure. If possible, please increase the font size, as it is difficult to read. Please include a brief descriptive legend. Please indicate the original sample of 22849 as those enrolled in the registry, and please clarify (here and or the first sentence of the results) why the number is different from the 22782 mentioned in the Results as enrolled in the Registry.

25. Figure 2: We suggest increasing the font size, if possible. Please also include a legend explaining the color map as it relates to low energy injury prevalence. Please note in the legend:

26. Figure 3: Please include a descriptive legend that describes what is illustrated in the figure.

27. Analysis Plan: We suggest renaming this file “S1 Analysis Plan” or similar. If possible, please note in the document that this describes a prospectively developed analysis plan for the project (as written it seems as if it is a preliminary report on the findings).

28. Supporting information files: Supplementary tables: Please provide a “clean” version of the document without markup. It was not clear which table went with which title/legend from tables 4-8. Please include the title and legend with each table individually.

29. Table 1: We suggest “no pre-existing medical conditions” or similar instead of “normal healthy patient” in the left column under Pre-Injury Health Status.

30. Supporting information files naming: You may use almost any description as the item name of your supporting information as long as it contains an "S" and number. For example, “S1 Appendix” and “S2 Appendix,” “S1 Table” and “S2 Table,” and so forth.

Please use whole numbers when naming your supporting information files. Combine separate parts (e.g., S1A and S1B Table) into one file (e.g. S1 Table) or rename with whole numbers (e.g., S1 and S2 Table).

Please match the names of your supporting information files with the supporting information captions within your manuscript. For example, a PDF file for “S2 Fig.” must be named “S2_fig.pdf”.

If you wish to refer to an element within a supporting information file, such as a table within a supporting text file, cite it in one of the following ways: “Table A in S1 Text,” “Table in S1 Table,” or “data in S1 Text.” Please do NOT cite it as “S1 Table in S1 Text.” This may lead to hyperlinking errors.

Comments from Reviewers:

Reviewer #2: We thank the authors for addressing the previous comments.

Reviewer #3: Markus Skrifvars, University of Helsinki

The authors have adressed my concerns and the paper has improved.

[LINK]

---

## [Editor Report · Decision Letter 4]

6 Aug 2021

Dear Dr Lecky, 

On behalf of my colleagues and the Academic Editor, Martin Schreiber, I am pleased to inform you that we have agreed to publish your manuscript "The burden of Traumatic Brain Injury from low energy falls among patients from 18 countries in the CENTER TBI Registry: A comparative cohort study." (PMEDICINE-D-20-05160R4) in PLOS Medicine.

Please also complete these remaining editorial requests:

-Abstract: Line 52: Please revise to “56 acute trauma receiving hospitals across 18 countries (17 countries in Europe and Israel)” to avoid confusion between distribution of countries and the numbers of acute trauma centers.

-Methods: Line 190: Please cite the ASA Physical Status Classification System guidelines as a reference (in the reference list).

PRESS

Sincerely, 

Caitlin Moyer, Ph.D. 

Associate Editor 

PLOS Medicine